# Liquid Biopsy and Circulating Biomarkers in Head and Neck Cancer: Advancing Non-Invasive Detection and Tailored Management

**DOI:** 10.3390/cancers17243974

**Published:** 2025-12-12

**Authors:** Ilaria Morelli, Chiara Ghirardini, Laura Faccani, Claudia Casanova, Ignacio Javier Fernandez, Stefano Tamberi

**Affiliations:** 1Oncology Unit, Santa Maria delle Croci Hospital-AUSL Romagna, 48121 Ravenna, Italy; chiara.ghirardini@auslromagna.it (C.G.); laura.faccani2@auslromagna.it (L.F.); claudia.casanova@auslromagna.it (C.C.); or stefano.tamberi@unibo.it (S.T.); 2Department of Medical Sciences (DIMEC), University of Bologna, 40126 Bologna, Italy; ignacio.fernandez4@unibo.it; 3Oncology Unit, University Hospital of Ferrara, 44124 Ferrara, Italy; 4Department of Otolaryngology Head and Neck Surgery, Santa Maria delle Croci Hospital-AUSL della Romagna, 48121 Ravenna, Italy

**Keywords:** head and neck squamous cell carcinoma, liquid biopsy, ctDNA, precision oncology

## Abstract

Head and neck squamous cell carcinoma (HNSCC) is a common and aggressive cancer with poor survival rates due to late diagnosis and biological heterogeneity. Liquid biopsy has emerged as a minimally invasive method to detect and monitor tumors using circulating biomarkers such as ctDNA, CTCs, cfRNA, miRNAs, extracellular vesicles and viral DNA. ctDNA, particularly viral ctDNA in HPV-positive cases, shows high sensitivity for detecting minimal residual disease and early relapse, often outperforming conventional imaging. Emerging biomarkers like TP53 mutations and methylation signatures hold promise for HPV-negative tumors. Despite technological advances in ddPCR and NGS, challenges remain in standardization and clinical validation. Integrating liquid biopsy into routine care could improve early detection, prognostic accuracy, and personalized treatment strategies.

## 1. Introduction

### 1.1. The Unmet Need for Biomarkers in Head and Neck Oncology

The incidence of Head and Neck cancer is rapidly growing, now representing the sixth most common non-skin cancer worldwide [1], with about 600,000 new cases diagnosed each year. The development of squamous cell carcinoma of the head and neck (HNSCC) is mainly related to long-term exposure to tobacco and alcohol, which may lead to widespread field cancerization of the upper aerodigestive tract. Over the past fifteen years, there has been a significant epidemi1ologic shift in the Western world, with a rise in human papillomavirus (HPV)-associated cancers [2], in particular for what concerns oropharyngeal cancer (OPC). More than 60% of SCCHN cases in the oral cavity, larynx and pharynx are diagnosed at locally advanced stage, requiring a comprehensive, multimodal treatment approach to achieve a cure.

Due to its high treatment burden and mortality, one of the major challenges in head and neck oncology is represented by the need of early diagnosis; symptoms arising from head and neck district are indeed often nonspecific (e.g., neck masses, nose bleeding, sore throat, swallowing problems, otalgia…), thus inevitably leading to delayed detection. Also, the absence of validated screening programs for general population may contribute to this delay. In addition, a distinct clinical challenge is represented by head and neck squamous cell carcinoma of unknown primary (SCCUP). This entity accounts for approximately 3–5% of all head and neck cancers and is characterized by cervical lymph node metastases without an identifiable primary tumor, despite extensive clinical and radiological work-up [3,4]. The absence of a detectable primary site complicates treatment decisions and may expose patients to unnecessarily extensive irradiation or surgery. In this context, novel biomarkers could potentially help to identify the site of origin, to refine patient stratification and to guide a more tailored therapeutic approach.

Disease monitoring as well may be challenging in this setting; post-treatment surveillance in head and neck cancer is based on regular clinical exams, most intensively in the first two years when >80% of recurrences occur [5]. Imaging is recommended at three months, with further scans performed only if symptoms or clinical findings suggest increased risk of recurrence. Given the poor prognosis after relapse, surveillance practices remain variable. Therefore, it is a matter of crucial importance to identify patients with minimal residual disease (MRD) before it is clinically or radiologically detectable and to resolve questions of equivocal disease status to permit earlier intervention and improve outcomes.

Finally, treatment management, especially for locally advanced disease, requires a highly multidisciplinary approach. Although tailored strategies based on tumor stage, HPV status and molecular features are improving outcomes, many patients experience significant morbidity from treatment-related toxicities. Balancing treatment intensity with quality of life remains a major unmet need.

Therefore, the aim of this narrative review is to provide an overview of the current evidence on the spreading adoption of liquid biopsy in head and neck cancers, with emphasis on their potential applications in early diagnosis, risk stratification, therapeutic decision-making and disease monitoring.

### 1.2. Liquid Biopsy and Biomarkers in Head and Neck Cancer: An Overview

Liquid biopsy is a term that encompasses the analysis of various bodily fluids, including blood, urine, cerebrospinal fluid and saliva [6]. In cancer diagnostics, it typically refers to assays that detect specific biomarkers. According to the National Cancer Institute, a liquid biopsy is “a test done on a sample of blood to look for cancer cells from a tumor that are circulating in the blood, or for pieces of DNA from tumor cells that are in the blood”. Compared to traditional tissue biopsy, liquid biopsy offers several advantages: it is minimally invasive, repeatable, cost-effective and feasible even in settings where surgical biopsy is risky [7,8]. Unlike tissue biopsy, it avoids sampling bias caused by tumor heterogeneity and does not require chemical preservation, providing fresh tumor material suitable for genomic, proteomic and metabolomic analyses. Additionally, it enables rapid and reliable real-time monitoring of cancer [7,8,9,10]. However, liquid biopsy is not yet a standard diagnostic tool and mainly complements tissue biopsy due to lower sensitivity and specificity, which can result in false positives or negatives [11,12]. Current assays also lack precision in identifying tumor origin in positive cases [13,14].

In head and neck cancer, several biomarkers can be detected through liquid biopsy, offering valuable insights into tumor burden, molecular alterations, treatment response and disease recurrence:Circulating tumor DNA (ctDNA): it consists of short fragments of tumor-derived DNA released into the bloodstream as cancer cells die or actively secrete genetic material. These fragments reflect the molecular characteristics of the tumor and can be analyzed using advanced sequencing technologies from a simple blood sample. ctDNA testing enables real-time assessment of tumor dynamics, detection of MRD and identification of emerging resistance mutations. While tissue biopsy remains essential for definitive diagnosis, ctDNA complements it by offering a non-invasive means of monitoring disease progression and guiding personalized therapy [15,16]Circulating tumor cells (CTCs): CTCs are malignant cells shed from the primary tumor or metastatic sites into the bloodstream, representing key intermediates in the metastatic process. Although most CTCs are rapidly destroyed, a small subset can survive and form distant metastases. Their detection in peripheral blood provides valuable prognostic and predictive information, as CTC counts correlate with disease progression, therapeutic response, and overall survival. Molecular characterization of CTCs allows exploration of tumor heterogeneity and resistance mechanisms. Despite technical challenges due to their rarity and biological variability, recent advances in enrichment and single-cell analysis technologies are improving their clinical utility [17,18,19].Cell-free RNA (cfRNA): cfRNA refers to extracellular RNA molecules found in various biological fluids, including plasma, serum, urine, cerebrospinal fluid, saliva, and pleural fluid [20]. Unlike intracellular RNA, cfRNA is released through active secretion or cell death processes such as apoptosis and necrosis [21]. As a non-invasive biomarker, cfRNA holds significant potential for liquid biopsy applications, supporting early diagnosis, disease monitoring, and prognosis in cancer and other pathological conditions [22,23,24,25].MicroRNAs (miRNAs): miRNAs are small, non-coding RNA molecules (19–25 nucleotides) that regulate gene expression at the post-transcriptional level. They are transcribed as primary miRNAs (pri-miRNAs), processed into precursor forms (pre-miRNAs), and finally matured to guide the RNA-induced silencing complex (RISC), which modulates target mRNA stability and translation. Dysregulated miRNA expression is frequently associated with cancer development and progression. Measuring circulating miRNA levels through liquid biopsy enables their use as sensitive and specific biomarkers for cancer diagnosis, prognosis, and treatment monitoring [26,27,28,29].Extracellular vesicles (EVs):EVs are nano-sized, membrane-bound particles secreted by cells under both physiological and pathological conditions. Present in all biological fluids, they contain DNA, RNA, proteins and lipids, reflecting the status of their cell of origin. Tumor-derived EVs play important roles in cancer progression by promoting metastasis, immune evasion, and chemoresistance. Moreover, EVs can be engineered as natural delivery systems for therapeutic molecules, offering promising applications in targeted cancer treatment [30,31,32,33].Viral nucleic acids: Detection of viral DNA or RNA in liquid biopsy samples—such as blood, urine or saliva—can indicate virus-associated malignancies. In particular, the identification of circulating HPV DNA (ctHPV) is highly relevant for HPV-related head and neck cancers. Measuring ctHPV levels allows for non-invasive monitoring of treatment response, detection of recurrence, and stratification of patients for therapy. With the application of next-generation sequencing (NGS), it is now possible to identify multiple HPV types and quantify viral load with high precision. Moreover, analyzing viral nucleic acids within extracellular vesicles may enhance detection sensitivity and stability [34,35,36,37].

The evaluation of ctDNA relies on several technological approaches, each offering specific strengths and limitations. The most widely used platforms include quantitative polymerase chain reaction (qPCR), droplet digital PCR (ddPCR) and NGS (Table 1). Detection technologies for circulating biomarkers other than ctDNA—including CTCs, EVs and RNA-based markers such as miRNAs and cfRNA—show considerably greater variability in analytical performance and cost. CTC detection platforms (e.g., immunomagnetic separation, microfluidic devices, or size-based enrichment) generally achieve moderate sensitivity and high specificity, but their performance is strongly influenced by tumor shedding and marker expression, while costs remain high due to specialized equipment and operator-dependent protocols. EV isolation methods—such as ultracentrifugation, size-exclusion chromatography, precipitation kits or immunoaffinity capture—tend to offer high specificity but variable sensitivity, largely dependent on the isolation workflow; cost and turnaround time range from moderate to high, especially for proteomic or RNA-sequencing downstream analyses. RNA-based biomarkers measured through qPCR, ddPCR, microarrays or RNA-seq generally achieve good analytical sensitivity, though specificity may be affected by contamination and normalization challenges; costs range from low–moderate for qPCR/ddPCR to high for sequencing platforms. Overall, compared with ctDNA detection technologies—where analytical performance is well characterized—methods for CTCs, EVs and RNA-derived biomarkers remain more heterogeneous, less standardized and often more resource-intensive, limiting direct comparability and current clinical applicability.

Meta-analytic evidence supports ddPCR as one of the most reliable techniques for ctDNA-based monitoring due to its high sensitivity, specificity and absolute quantification capability. qPCR is more widely available and cost-efficient but lacks the sensitivity needed for detecting ctDNA at very low levels. Conversely, NGS technologies enable extensive genomic characterization of tumors, offering insights into tumor heterogeneity and treatment resistance, although they are associated with higher financial and logistical demands.

In summary, ddPCR is particularly suited for real-time clinical surveillance, whereas NGS serves as the preferred strategy for extensive mutation profiling and precision-oncology applications.

The choice of technology should depend on the clinical scenario, balancing sensitivity requirements with economic and infrastructural considerations:


**Application**

**Preferred Approach**
MRD trackingddPCRBroad biomarker discovery and resistance mechanismsHybrid-capture NGSTargeted mutation verificationAmplicon-based NGS or ddPCR

Although qPCR, ddPCR and NGS represent the most widely adopted approaches for ctDNA analysis, they are also extensively applied to other circulating biomarkers, including cfRNA, miRNAs, viral nucleic acids and nucleic acids extracted from EVs. These technologies enable both targeted and comprehensive molecular profiling, supporting the detection of minimal residual disease and real-time evaluation of tumor evolution. Conversely, CTC analysis additionally requires cell-based enrichment and characterization methods, such as immunomagnetic separation, flow cytometry and imaging approaches, due to their extreme rarity and cellular heterogeneity. Overall, the complementary use of molecular and cellular platforms enhances the potential of liquid biopsy to capture tumor heterogeneity and improve diagnostic, prognostic and therapeutic decision-making.

Building on this overview, the purpose of our work is to delve into circulating biomarkers detectable via liquid biopsy, highlighting the current clinical and translational evidence and their potential impact on early diagnosis, prognosis, treatment monitoring and response assessment and personalized management of head and neck cancers.

A comprehensive literature search was performed in PubMed, Embase and Scopus using the following search string: (ctDNA OR “circulating tumor DNA” OR “liquid biopsy”) AND (“head and neck cancer” OR “head and neck neoplasm” OR “head and neck tumor” OR HNSCC OR “head and neck squamous cell carcinoma”). We included English-language articles published up to 2025 that reported on circulating biomarkers in HNSCC and addressed diagnosis, prognosis or therapeutic monitoring. Exclusion criteria were non-English publications, conference abstracts without full text, studies not related to HNSCC, and studies limited to animal models or in vitro experiments. While this review remains narrative in nature, these measures were implemented to ensure a transparent and reproducible study selection process.

## 2. Clinical Applications of Biomarkers in Head and Neck Cancer

### 2.1. Early Detection of Head and Neck Cancers Using Circulating Biomarkers

The diagnosis of HNC is currently based on imaging procedures such as magnetic resonance (MRI), computed tomography (CT) and positron emission tomography/computed tomography (PET/CT), supplemented by histopathological analysis. However, obtaining a tissue biopsy can sometimes be challenging and technically difficult due to surgical complications (e.g., bleeding in those with hemorrhagic diathesis) and the sample may not be sufficient for analysis, this making it necessary to repeat the sampling in such cases. Liquid biopsy, on the other hand, is an easily accessible and less invasive test for the patient. Furthermore, current diagnostic methods have limited value in early diagnosis, which is why most patients present with advanced disease at diagnosis. This has led to considerable interest in the potential application of ctDNA in the screening and early diagnosis of HNCs in order to improve patient survival (Figure 1).

#### 2.1.1. Choosing the Right Fluids for ctDNA Detection

Current studies on ctDNA in HNSCC have mainly focused on blood (plasma or serum) and saliva as biological sources. Saliva represents a unique and promising biofluid for biomarker detection, as it is easily accessible, non-invasive and can reflect molecular alterations associated with head and neck carcinogenesis [39]. In a pivotal study by Wang et al. involving 93 patients with HNSCC, both plasma and saliva samples were analyzed to identify somatic mutations (TP53, PIK3CA, CDKN2A, HRAS, NRAS) and HPV serotypes (HPV-16, -18) [40]. The authors found that the detection rate of tumor DNA varied by tumor site and sample type. Specifically, tumor DNA was identified in 100% of oral cavity, 91% of oropharyngeal and 100% of both laryngeal and hypopharyngeal cancer patients. In saliva, ctDNA was detectable in all patients with oral cavity tumors but in only 47–70% of those with tumors from other head and neck subsites. Conversely, plasma showed higher positivity rates for non-oral cavity cancers (86–100%) but slightly lower for oral cavity cases (80%).

These findings suggest that saliva is preferentially enriched for tumor DNA originating from oral cavity lesions, whereas plasma better reflects tumors arising from deeper head and neck sites. Importantly, in a subgroup of 43 patients, combining plasma and saliva testing significantly improved overall detection sensitivity across tumor sites [40]. Subsequent investigations have confirmed the value of a multi-fluid approach. Ahn et al. demonstrated that integrating plasma and saliva analyses enhanced surveillance performance in HPV-related HNSCC [41], while Bu et al. further expanded this concept through a *tri-modal liquid biopsy* approach that simultaneously analyzed circulating tumor cells, exosomes and cell free DNA (cfDNA) using a machine learning algorithm [42]. With recent advances in molecular diagnostics, blood-based genetic biomarkers—particularly ctDNA—are gaining increasing relevance across multiple cancer types. Circulating tumor DNA fragments, released into the bloodstream through apoptosis and necrosis of tumor cells, carry tumor-specific genetic alterations that can be exploited for minimally invasive cancer detection and monitoring. Liquid biopsy offers a minimally invasive window into the molecular landscape of head and neck cancers. By analyzing ctDNA, CTCs or other tumor-derived components in blood or saliva, this approach can reveal a variety of clinically relevant information, including viral etiologies (such as HPV or Epstein–Barr Virus (EBV), tumor-specific genetic alterations (e.g., TP53, PIK3CA, HRAS) and markers of disease burden and/or treatment response. This makes liquid biopsy a powerful tool not only for early detection and diagnosis, but also for monitoring disease progression, recurrence, and response to therapy.

#### 2.1.2. Use of ctDNA in HPV-Positive Disease (Table 2)

Several studies in the literature reported high sensitivity and specificity in the detection of ctDNA in HPV positive OPSCC, ranging from 91.5 to 98.4% and 98.6 to 100%, respectively, with positive and negative predictive values above 90% [43,44,45,46,47,48,49,50]. The above-mentioned evidence demonstrates that liquid biopsy techniques can be a useful complement to traditional diagnostic pathways, as they can accelerate diagnosis when referral times to an otolaryngologist are long or when patient comorbidities preclude general anesthesia or require additional cardiopulmonary clearance for general anesthesia. Complications of operative laryngoscopy, although rare, include anesthetic sequelae, therapeutic errors, dental injuries and esophageal perforations. Furthermore, tissue biopsy does not always guarantee an accurate diagnosis, as false-negative results may occur in cases of low-volume or submucosal disease due to sampling errors.

In general, liquid biopsy techniques for early cancer diagnosis show low concentrations of ctDNA in body fluids, especially in early stages of the disease. However, a study conducted by Rettig et al. [51] showed that HPV ctDNA could be detected in plasma samples approximately 30.5 months prior to diagnosis, with no false positives. Further analysis in a secondary cohort showed that HPV ctDNA was detected in the pre-diagnostic plasma of 43% of HPV-positive OPC patients [51]. Although promising, conclusions from these findings should be drawn with caution as these studies had small sample sizes and therefore require further validation. In a prospective observational study by Siravegna et al. [48] a total of 61 patients with newly diagnosed or suspected untreated HNSCC were enrolled, as well as 70 HPV-negative controls. All patients with HNSCC underwent a standard clinical examination, which included fine needle aspiration and/or tissue biopsy of the primary tumor. The diagnostic success rate of the first diagnostic attempt was 72%, with 28% of patients requiring a second diagnostic attempt with tumor biopsy to determine the diagnosis. In contrast, detection of serum HPV ctDNA for the diagnosis of HPV-positive HNSCC showed a sensitivity of 98.4%, a specificity of 98.6%, a positive predictive value (PPV) of 98.4%, and a negative predictive value (NPV) of 98.6%, thus demonstrating that HPV ctDNA detection can offer a non-invasive diagnostic approach for HPV-positive HNSCC with greater accuracy and, above all, reduced diagnosis times. HPV ctDNA was also found to have high sensitivity, even in a cohort with low disease burden (75% of patients with stage I OPSCC), thus increasing its appeal as a screening tool. Even in a subgroup analysis of a study conducted by Wang Y. et al., tumor DNA was identified in 100% (n = 10) of patients with early disease (stage I and II) and in 95% (n = 37) of patients with advanced disease (stage III and IV). In addition, tumor DNA was detected in all patients with lesions of the oral cavity (n = 15), larynx (n = 7) and hypopharynx (n = 3) and in 91% (n = 22) of patients with oropharyngeal tumors [40]. In a study conducted by Damerla et al. [43] on 97 patients with locoregionally confined oropharyngeal squamous cell carcinoma, HPV ctDNA analysis showed high sensitivity (95.6%) and specificity (100%) rates, demonstrating the usefulness of ctDNA even in early-stage disease. It is interesting to note that tumor DNA appears to be more detectable in early-stage HNSCC than in other early-stage cancers. It is unclear whether this is due to different techniques used in the conducted studies or perhaps to anatomical considerations regarding head and neck tumors, such as increased vascularization of the region.

When interpreting circulating HPV DNA results, it is important to consider that the mere presence of HPV infection does not always indicate malignant transformation. In this context, another factor to consider is that HPV infections, in addition to causing head and neck cancers, can also cause non-invasive lesions associated with HPV or HPV infection without carcinogenesis. In this regard, Jeannot et al. did not detect circulating HPV DNA in 18 patients with cervical intraepithelial neoplasia [52]. However, this was a small study, and it remains possible that patients with HPV infection but without cancer may release HPV DNA into their plasma, albeit at levels below the detection limit of current methods.

**Table 2 cancers-17-03974-t002:** Summary of studies on circulating HPV DNA as a biomarker in Head and Neck Squamous Cell Carcinoma (HNSCC).

First Author, Year	Setting	Biomarker and Assay Type	N. Patients	Main Findings	Clinical Implications
**Mijares K., 2024 [47]**	Clinical surveillance and diagnostic evaluation of HPV + HNSCC	Circulating tumor HPV DNA (ctHPVDNA) using ddPCR	167 total (141 HPV+)	Detectable ctHPVDNA in 94.3% of HPV+ tumors; sensitivity 91.7%, specificity 100%, PPV 100%, NPV 63.6%. Rare non-HPV16 genotypes explain most false negatives.	Strong diagnostic and surveillance performance; highly reliable for confirming HPV+ disease; negative result requires caution due to limited NPV, especially for non-HPV16 tumors
**Rettig EM, 2022 [51]**	Case–control study using archived plasma from biobank	ctHPV16DNA detection via digital droplet PCR (TTMV assay)	12 cases + 100 controls	ctHPV16DNA detectable in 30% of HPV16 + HNSCC patients (43% in HPV16+ OPSCC) up to 19–43 months before diagnosis; not detected in controls	Supports ctHPV16DNA as a potential biomarker for preclinical detection of HPV-positive HNSCC
**Sastre-Garau X., 2021 [50]**	Prospective, multicenter (France & Senegal)	ctHPV DNA via NGS-based CaptHPV assay (broad genotype capture)	135 total (80 HPV-positive tumors; multiple primary sites including 25 OPSCC)	Sensitivity 95% and specificity 98.1% for detecting HPV-positive carcinomas from plasma; 15 different HPV genotypes detected	Blood-based diagnostic capable of detecting any HPV genotype, enabling personalized tumor markers and expanding ctDNA testing to non-HPV16 cancers
**Siravegna G., 2021 [48]**	Prospective observational study comparing noninvasive vs. standard diagnostic workup for HPV + HNSCC	Plasma ctHPVDNA measured with custom ddPCR targeting HPV genotypes 16/18/33/35/45	140 (70 cases, 70 controls)	ctHPVDNA alone: sensitivity 98.4%, specificity 98.6%; Combined ctHPVDNA + imaging: higher diagnostic accuracy vs. standard care, 36–38% lower cost, diagnosis 26 days faster	ctHPVDNA-based noninvasive diagnosis improves accuracy, reduces costs and accelerates diagnosis, serving as a promising alternative to tissue biopsy-based diagnostic pathways
**Damerla RR, 2019 [43]**	Prospective plasma-based diagnostic cohort in early-stage HPV-associated cancers	Plasma HPV16/33 ctDNA by droplet digital PCR (ddPCR)	105 (97 OPSCC, 8 anal SCC) + 27 controls	Overall sensitivity 95.6% and specificity 100%. Detectable ctDNA in all low-volume HPV + OPSCC cases	ddPCR ctHPV-DNA is highly accurate for early detection and treatment monitoring in HPV-associated cancers; supports potential use in screening
**Wang Y., 2015 [40]**	Prospective sample collection (plasma + saliva) in HNSCC	Tumor-specific DNA detection (somatic mutations & HPV DNA), qPCR/sequencing-based assays	93	Detectable tumor DNA in 96% of patients when combining saliva + plasma; early-stage detection 100%; saliva highly sensitive for oral cavity tumors, plasma for non-oral cavity sites; postsurgical detection of recurrence before clinical diagnosis in 3 cases	Combined saliva + plasma sampling enhances detection across sites and may enable early recurrence detection and screening in HNSCC

**HNSCC**, Head and neck squamous cell carcinoma; **HPV**, Human papillomavirus; **ctDNA**, Circulating tumor DNA; **ctHPVDNA**, Circulating tumor HPV DNA; **ddPCR**, Droplet digital polymerase chain reaction; **NPV**, Negative predictive value; **PPV**, Positive predictive value; **NGS**, Next-generation sequencing; **OPSCC**, Oropharyngeal squamous cell carcinoma; **qPCR**, Quantitative polymerase chain reaction.

#### 2.1.3. EBV-Associated ctDNA (Table 3)

In addition to the well-established link between HPV and HNSCC, EBV is strongly associated with nasopharyngeal carcinoma (NPC). Large-scale screening studies have demonstrated that EBV ctDNA analysis is useful not only for early detection of primary NPC [53,54] but also for identifying early recurrence during post-treatment surveillance. In a prospective study, Lo et al. reported high EBV ctDNA levels in 55/57 patients (96%) with NPC compared to 3/43 controls (7%), highlighting the value of plasma EBV ctDNA as a screening biomarker [55]. Other large prospective investigations in endemic regions confirmed promising sensitivity (86.8–97.1%) and specificity (90–98.6%) for EBV ctDNA in NPC screening [53,56]. Similar findings were observed in a non-endemic population by Krishna et al., although sensitivity was lower (75%) [57]. Several other EBV-related biomarkers have been explored, including viral capsid antigen and early antigen IgA serology, but results have been inconsistent, warranting further evaluation for early HNC detection [58,59,60]. Modeling studies by Miller et al. suggested that combining EBV ctDNA with serology could be a cost-effective strategy, potentially increasing 10-year survival from 71% to 86.3% [61]. Beyond early detection, EBV ctDNA also has prognostic value. Wei et al. [62] showed that patients with consistently negative plasma EBV DNA were more likely to present with earlier T and N stages and had significantly better 3-year disease-free survival (95% vs. 84.4%), distant metastases-free survival (98.3% vs. 89.4%) and overall survival (100% vs. 97.6%) compared with EBV-positive patients, suggesting its utility as a prognostic biomarker in addition to screening.

**Table 3 cancers-17-03974-t003:** Summary of studies on plasma EBV DNA and serology for early detection and prognosis of Nasopharyngeal Carcinoma (NPC).

First Author, Year	Setting	Biomarker and Assay Type	N. Patients	Main Findings	Clinical Implications
**Chen WJ, 2021 [54]**	EBV-seropositive population in NPC high-risk region	Plasma EBV DNA (real-time qPCR)	1363 followed, 30 NPC cases	Higher plasma EBV DNA strongly predicted NPC development; HR up to 39.79 for ≥1000 copies/mL; predictive value dropped when excluding cases within 3–4 years	EBV DNA improves NPC early detection, enhances serology-based screening; best predictive value within 3-year window, useful for surveillance of high-risk individuals
**Wei ZG, 2021 [62]**	Single-center cohort (NPC patients)	Plasma EBV DNA, quantitative assay	480 NPC patients	Patients with consistently negative EBV DNA had earlier T and N stage; significantly better 3-year DFS (95.0% vs. 84.4%), DMFS (98.3% vs. 89.4%), and OS (100% vs. 97.6%) compared with EBV-positive patients	Consistently negative EBV DNA identifies NPC patients with earlier clinical stage and superior survival outcomes; may help in prognostication and risk stratification
**Tay JK, 2020 [60]**	Prospective cohort of individuals with first-degree family history of NPC	EBV-EA IgA & VCA IgA serology + serum cf-EBV DNA	524 high-risk individuals (5 NPC cases)	Screening detected NPC in 0.96% (199/100,000 person-years), 80% T1 stage at diagnosis; EBV-EA IgA showed 94.6% specificity & 15.2% PPV, outperforming VCA IgA and cf-EBV DNA; rising EA IgA titers preceded diagnosis	Screening in familial-risk groups can detect early, asymptomatic NPC; EBV-EA IgA adds diagnostic value for biopsy triage and surveillance
**Chan KCA, 2017 [53]**	Prospective population-based screening	Plasma EBV DNA, real-time PCR	20,174 asymptomatic participants	EBV DNA detectable in 5.5% of participants; 34 NPC cases detected among 309 persistently positive individuals; sensitivity 97.1%, specificity 98.6%; 71% of detected NPC were stage I/II vs. 20% in historical cohort; superior 3-year progression-free survival (97% vs. 70%)	Plasma EBV DNA is effective for early detection of asymptomatic NPC, enabling diagnosis at earlier stages and improving patient outcomes
**Ji MF, 2014 [56]**	Population-based NPC screening	Plasma EBV DNA, real-time PCR; previously identified high-risk participants via VCA/IgA and EBNA1/IgA	825 high-risk participants	Using 0 copies/mL as cutoff: sensitivity 86.8% for NPC detected within 1 year; PPV 30%, NPV 99.3%; lower sensitivity for early stage NPC (81.5%) vs. advanced NPC (100%); for NPC detected after 1 year, baseline positivity 50%	Plasma EBV DNA improves diagnostic accuracy in high-risk individuals, but limited value for early-stage NPC detection and prediction of NPC development

**EBV**, Epstein–Barr virus; **NPC**, Nasopharyngeal carcinoma; **qPCR**, Quantitative polymerase chain reaction; **cf-EBV DNA**, Cell-free EBV DNA; **EA IgA**, EBV early antigen immunoglobulin A; **VCA IgA**, EBV viral capsid antigen immunoglobulin A; **DFS**, Disease-free survival; **DMFS**, Distant Metastasis-free Survival; **OS**, Overall Survival; **PPV**, Positive Predictive Value; **NPV**, Negative Predictive Value.

#### 2.1.4. Genomic Alterations Detected by ctDNA in HNSCC

HPV-positive and HPV-negative HNSCCs differ in terms of commonly identified oncogenic mutations, which could serve as identifiable targets for liquid biopsy. Data from The Cancer Genome Atlas (TCGA) indicated that PIK3CA was the most commonly mutated oncogene in HPV-positive HNSCC, while genomic alterations in HPV-negative HNSCC were mostly limited to tumor suppressor genes, including TP53, NOTCH1 and FAT1 mutations and CDKN2A inactivation [63,64,65]. An analysis by Porter et al. of ctDNA from 60 patients with HNSCC showed that TP53 (68%), PIK3CA (34%), NOTCH1 (20%), and ARID1A (15%) were the most frequent mutations, consistent with tumor sequencing data showing that TP53 (48%) and PIK3CA (24%) were the most common mutations [66].

The detection of TP53 mutations in ctDNA is also an important prognostic factor, as it is linked to the presence of regional metastases and reduced PFS and OS, while no prognostic value was found for CDKN2A and NOTCH1 mutations in ctDNA [67].

Abnormal DNA methylation such as global hypomethylation, regional hypermethylation at different genomic locations (mainly CpG islands), and direct mutagenesis in methylated cytosines are all factors contributing to carcinogenesis and tumor progression. Over the past decade, epigenetic alterations, primarily aberrant DNA methylation, have been shown to play a significant role in HNSCC [68,69].

The study of these changes in DNA methylation is important because it has been shown that they can be detected in plasma up to four years before a conventional diagnosis, thus representing a useful element in early diagnosis [70].

Misawa et al. [71] demonstrated in particular that methylation of the CALML5, DNAJC5G and LY6D genes in ctDNA was highly effective in differentiating between patients with HPV-positive oropharyngeal SCC and healthy controls. EDNRB hypermethylation in ctDNA, on the other hand, significantly associated with HNSCC when comparing an aggregate group of HPV-positive and HPV-negative HNSCC patients with healthy controls, but it is detectable in only a minority of HNSCC patients, limiting its value as a diagnostic biomarker [72].

With regard to ctDNA methylation in saliva samples, Lim et al. found that the RASSF1α, CDKN2A, TIMP3, and PCQAP/MED15 genes had higher methylation levels in HPV-negative HNSCC patients than in healthy controls, while the same genes had lower methylation levels in HPV-positive HNSCC patients than in healthy controls [73].

Serial liquid biopsy testing of patients showed that genomic variant frequencies and intratumoral heterogeneity levels changed before and at the time of HNSCC recurrence, confirming, as per previous evidence, the ability of ctDNA to track temporal clonal evolution and identify dynamic changes in ctDNA that predict recurrence/metastasis before clinical detection [74,75].

#### 2.1.5. Limitation of ctDNA Adoption in Early Diagnosis

Despite the promising diagnostic characteristics of these emerging biomarkers, their implementation for screening and early diagnosis of HNSCC is limited due to the relatively low prevalence of the disease in the general population, despite its increasing incidence. As a result, the positive predictive value of even a near-perfect biomarker remains low, with a large number of screenings required to detect a single cancer [76]. However, one specific population of particular interest for whom ctDNA analysis may have reasonably high predictive value and facilitate early diagnosis is the subset of patients presenting with suggestive signs and symptoms (sore throat, tonsillar asymmetry, or neck swelling) that may be related to undiagnosed HNSCC [77].

It should be noted that ctHPVDNA can also be detected in other HPV-positive malignant neoplasms that are not oropharyngeal, including SCCs arising from other head and neck subsites, such as sinonasal, anal and cervical [78,79]. Furthermore, while cervical cancer has an identifiable precursor lesion for screening, nothing similar has been described for OPSCC.

### 2.2. Risk Assessment and Prognostic Insights from Liquid Biopsy Biomarkers (Table 4)

Several studies have consistently shown that elevated ctDNA and cfDNA levels are strongly linked to advanced disease stages, reflecting higher tumor burden, lymph node involvement and overall clinical stage (Table 4).

Following this evidence, an Asian cohort study [80] demonstrated the diagnostic and prognostic relevance of CTCs in oral cancer patients: levels were significantly higher in patients than in healthy controls and correlated with clinical stage, differentiation and nodal metastasis, highlighting their value as indicators of tumor aggressiveness.

Concordant findings emerged from a recent Spanish multicenter study [81], which reported higher baseline plasma cfDNA concentrations in HNSCC patients compared to controls, including those with early-stage disease, thus supporting its role as a minimally invasive diagnostic biomarker.

A complementary study [82] employing whole-exome sequencing of cfDNA from 50 oral squamous cell carcinoma (OSCC) patients revealed that plasma mutation burden was significantly correlated with clinical stage and the presence of distant metastasis. Frequently mutated genes included *TTN*, *PLEC*, *SYNE1* and *USH2A*, while canonical drivers such as *KMT2D* and *LRP1B* were also enriched in advanced cases. These findings reinforce the association between genomic complexity in cfDNA and more aggressive clinical behavior, supporting its role in stage-related risk assessment.

In addition, a single-center study [83] evaluated genome-wide copy number aberrations in cfDNA from 116 HNSCC patients and derived a copy number instability (CNI) score. Higher CNI values were associated with advanced tumor stage, lymph node involvement and poorer overall survival, outperforming traditional clinical features as a prognostic predictor. These findings suggest that ctDNA-based genomic instability metrics may provide a quantitative, minimally invasive tool for risk stratification and pre-treatment therapeutic planning.

In parallel, metabolic imaging studies have demonstrated that high FDG-PET tumor burden correlates with positive liquid biopsy and elevated variant allele frequency (VAF) [84]. Collectively, these data highlight the complementary role of genomic and metabolic signatures in disease stratification and pre-treatment evaluation in HNSCC.

#### 2.2.1. Risk Stratification in HPV-Positive Cohort

In HPV-positive OPSCC, ctHPV-DNA or tumor tissue–modified viral (TTMV)-HPV DNA has emerged as a valuable biomarker closely reflecting radiologic and clinical disease status. Beyond its diagnostic and monitoring applications, quantitative assessment of ctHPV16 viral load has also shown prognostic relevance. In a cohort of 91 patients with HPV16-positive OPSCC treated with chemoradiotherapy [85], higher pre-treatment ctHPV16 DNA levels measured by qPCR were significantly associated with an increased risk of distant metastasis and inferior metastasis-free survival. Patients with elevated viral load exhibited more than a twofold higher risk of metastatic progression, supporting its potential role in pre-treatment risk stratification and outcome prediction.

Pretreatment data have shown that higher plasma TTMV-HPV DNA levels correlate with more advanced stage and greater nodal burden on PET-CT imaging, underscoring the association between cfDNA load and overall tumor volume [86]. Following therapy, serial ctHPV-DNA monitoring provides a highly sensitive and specific means of detecting recurrence. Persistent or rising ctDNA levels strongly predict disease relapse and frequently precede radiologic evidence of recurrence, whereas ctDNA clearance aligns with imaging-based remission. In a cohort of 34 patients with HPV-driven OPSCC, a strong positive correlation was observed between ctDNA kinetics and imaging findings in recurrent cases, while undetectable ctHPV-DNA was invariably associated with absence of relapse, corresponding to a 100% negative predictive value [87].

**Table 4 cancers-17-03974-t004:** Summary of studies on ctDNA-based risk stratification in Head and Neck Cancer, including HPV-positive Oropharyngeal Squamous Cell Carcinoma.

First Author, Year	Setting	Biomarker and Assay Type	N. Patients	Main Findings	Clinical Implications
**Rodriguez-Ces AM, 2025 [81]**	Multicentre, prospective study in patients with HNSCC across all stages	ccfDNA quantified by fluorometry (Qubit) and qPCR	85 HNSCC patients, 28 healthy controls	Baseline plasma ccfDNA significantly higher in HNSCC vs. controls (AUC = 0.705); elevated levels even in early-stage disease; lower post-treatment ccfDNA associated with longer PFS (16.37 vs. 9.63 months, *p* < 0.05); high inter-patient variability in ccfDNA kinetics	Fluorometric ccfDNA quantification shows promise as a minimally invasive biomarker for prognosis in HNSCC, warranting validation in larger studies
**Cooke PV, 2025 [86]**	Single tertiary-center cohort study of patients with HPV-positive OPSCC	(TTMV) HPV DNA quantified by fragment count (fragments/mL)	203 HPV+ OPSCC patients	Higher pretreatment TTMV-HPV DNA levels associated with advanced clinical T and N stage (aOR for cT3/4 = 2.51; aOR for cN1 = 4.26; cN2/3 = 3.64) and greater total tumor plus nodal volume on PET-CT (aOR = 1.04); no significant association with histopathologic risk factors or survival outcomes	Pretreatment TTMV-HPV DNA load reflects overall tumor and nodal burden, supporting its potential use as a noninvasive indicator of disease extent and a tool for refining treatment intensity in HPV+ OPSCC
**Agarwal A., 2024 [87]**	Cohort study in patients with HPV-driven OPSCC undergoing posttreatment surveillance	ctHPV-DNA quantified from plasma	34 HPV-positive OPSCC patients with ≥3 sequential imaging and ctDNA assessments	Strong positive correlation between ctHPV-DNA levels and imaging findings in recurrent cases; ctHPV-DNA test showed 100% negative predictive value for ruling out recurrence	ctHPV-DNA assay provides a highly sensitive and specific, noninvasive tool for posttreatment surveillance in HPV-driven OPSCC, enabling earlier detection of recurrence and supporting integration into routine multidisciplinary management
**Mazurek AM, 2024 [85]**	Prospective cohort study of patients with HPV16-positive OPSCC treated with CRT	ctHPV16 quantified by qPCR	91 ctHPV16-positive OPSCC patients	Higher pre-treatment ctHPV16 viral load significantly associated with increased risk of distant metastasis (VL 4.09 vs. 3.25; *p* = 0.009); HR for MFS = 2.22 (*p* = 0.015); optimal cutoff VL = 3.556; 5-year LRFS, MFS, and OS were 88%, 90%, and 81%, respectively	Elevated baseline ctHPV16 viral load identifies patients at higher risk of metastatic progression and poorer outcomes; supports use of ctHPV16 VL for pre-treatment risk stratification in HPV-driven OPSCC
**Lin LH, 2023 [82]**	Comprehensive genomic profiling study in OSCC	cfDNA analyzed by WES with multiple variant calling pipelines and IGV validation	50 paired plasma and whole blood samples	Plasma cfDNA mutation burden significantly correlated with clinical stage and distant metastasis; recurrently mutated genes included TTN, PLEC, SYNE1, USH2A, and known drivers (KMT2D, LRP1B, TRRAP, FLNA)	cfDNA WES reveals key genomic alterations associated with OSCC progression and metastasis, supporting its use for prognostic stratification and identification of actionable therapeutic targets
**Silvoniemi A., 2023 [84]**	Prospective study correlating ctDNA with FDG-PET/CT metabolic parameters in HNSCC	ctDNA analyzed by NGS; VAF correlated with FDG-PET/CT metrics	26 HNSCC patients	Maximum VAF in ctDNA correlated positively with metabolic tumor volume (r = 0.510, *p* = 0.008) and total lesion glycolysis (r = 0.584, *p* = 0.002); ctDNA positivity associated with high WB-TLG; concordant ctDNA/tissue variants also linked to metabolic burden	ctDNA levels and metabolic imaging provide complementary prognostic information; integrating genomic and metabolic markers may enhance pre-treatment risk stratification and therapeutic planning in HNSCC
**Schirmer AM, 2018 [83]**	Single-center, non interventional observational study in HNSCC patients	Cell-free tumor DNA CNAs assessed by low-coverage NGS; genome-wide CNI score derived	116 HNSCC patients (103 presurgery samples) and 142 tumor-free controls	High CNI values strongly associated with lymph node involvement and poorer overall survival (HR 4.89, *p* = 0.01); AUC for tumor detection 87.2%; CNI outperformed conventional staging features as prognostic predictor	CNI score represents a robust, minimally invasive biomarker for predicting lymph node involvement and survival in HNSCC, supporting its potential integration into pre-treatment risk stratification and treatment planning

**ccfDNA**, Circulating Cell-free DNA; **Qubit**, Fluorometric quantification method; **qPCR**, Quantitative Polymerase Chain Reaction; **HNSCC**, Head and Neck Squamous Cell Carcinoma; **PFS**, Progression-Free Survival; **AUC**, Area Under the Curve; **TTMV-HPV DNA**: Tumor Tissue–Modified Viral Human Papillomavirus DNA; **OPSCC**, Oropharyngeal Squamous Cell Carcinoma; **PET-CT**, Positron Emission Tomography–Computed Tomography; **aOR**, Adjusted Odds Ratio; **ctHPV-DNA**, Circulating Tumor Human Papillomavirus DNA; **CRT**, Chemoradiotherapy; **ctHPV16**, Circulating Tumor Human Papillomavirus type 16 DNA; **VL**, Viral Load; **MFS**, Metastasis-Free Survival; **LRFS**, Locoregional Recurrence-Free Survival; **OS**, Overall Survival; **cfDNA**, Cell-free DNA; **WES**, Whole-Exome Sequencing; **IGV**, Integrative Genomics Viewer; **NGS**, Next-Generation Sequencing; **VAF**, Variant Allele Frequency; **FDG-PET/CT**, Fluorodeoxyglucose Positron Emission Tomography/Computed Tomography; **WB**, Whole Body; **TLG**, Total Lesion Glycolysis; **CNI**, Copy Number Instability; **HR**, Hazard Ratio.

#### 2.2.2. Prognostic Significance of Specific Genetic Alterations Detected in ctDNA

The prognostic assessment of HNSCC is increasingly enriched by liquid biopsy approaches, with ctDNA emerging as a minimally invasive tool that captures tumor-specific genetic and epigenetic alterations in real time. Acting as a dynamic biomarker of tumor burden and molecular evolution, ctDNA provides critical insights into disease progression and patient outcomes. Evidence from recent meta-analyses supports the robust prognostic value of ctDNA in HNSCC. A meta-analysis of eight studies including 886 HPV-negative patients [88] demonstrated a consistent association between ctDNA alterations and poorer survival, particularly highlighting TP53 mutations and methylation of SEPT9 and SHOX2 as markers of reduced overall, disease-free and progression-free survival. In line with these results, a broader meta-analysis [89] encompassing 22 studies with over 5000 HNSCC patients confirmed that positive ctDNA detection or aberrant methylation patterns is significantly associated with worse overall and progression-/recurrence-free survival (pooled HR = 2.00 for OS, HR = 3.54 for PFS/RFS), reinforcing the utility of ctDNA as a reliable prognostic biomarker and emphasizing that patients lacking detectable ctDNA or methylation alterations generally experience more favorable outcomes.

Prospective studies further support these findings: in a cohort of 62 patients [90] pathogenic variants in TP53, CDKN2A, HRAS and PIK3CA detected in tumor or circulating DNA were associated with worse overall survival, while another study focusing on TP53-mutated ctDNA [91] confirmed its link with advanced nodal involvement and shorter progression-free survival, underscoring TP53 as a pivotal determinant of poor prognosis.

Similarly, a recent large-scale retrospective review of 75 HNSCC patients [67] demonstrated that ctDNA alterations—particularly in DNA repair genes and TP53—were significantly associated with decreased overall survival and with disease presence and extent at last follow-up. Concordance between tumor tissue DNA and ctDNA was limited, but actionable ctDNA alterations were identified in approximately two thirds of patients, highlighting both the prognostic and potential precision medicine relevance of ctDNA profiling.

In this context, smaller studies [92] have further illustrated the prognostic relevance of ctDNA-based detection of specific genetic alterations. In a cohort of patients with oropharyngeal squamous cell carcinoma (OPSCC), matched somatic variants identified in both tumor tissue and plasma cfDNA—mainly involving *TP53* and *FBXW7*—were observed exclusively in nonresponders or patients with persistent disease. The concordant detection of mutations such as *TP53* G325fs, R282W, R273C, and *FBXW7* R505G/L, although at low allele frequencies in cfDNA, indicated a potential association with disease recurrence or treatment resistance. Variant detection was more frequent in HPV-negative nonresponders, suggesting that cfDNA genotyping may help characterize biologically more aggressive disease subsets.

Similarly, a small cohort of 38 patients with inoperable HNSCC treated with chemoradiotherapy [93] showed that *TP53* mutations—either persistent or newly emerging post-treatment—were associated with advanced T stage and poorer outcomes. Detectable mutations before or after CRT predicted shorter locoregional recurrence-free and distant metastasis-free survival, underscoring the prognostic value of serial ctDNA monitoring to identify patients at high risk of recurrence or progression.

Beyond these molecular alterations, ctDNA has also proven useful in estimating recurrence risk. Evidence indicates that patients with detectable ctDNA at baseline or during follow-up exhibit a higher likelihood of disease relapse and cancer-related mortality, even prior to clinical or radiologic detection. This property allows ctDNA to inform pre-treatment risk stratification and guide individualized therapeutic decisions, complementing conventional clinicopathological prognostic factors [94].

Complementing ctDNA, CTCs and circulating tumor microemboli (CTM) [95] provide additional prognostic and potential predictive value. High CTC counts and CTM positivity are associated with poorer survival and may identify patients more likely to benefit from induction chemotherapy. Extending this evidence, molecular characterization of CTCs has also revealed the prognostic significance of PD-L1 expression. In a prospective cohort of patients with locally advanced HNSCC treated with curative intent [96], PD-L1 overexpression in EpCAM(+) CTCs was detected in approximately one quarter of cases and was independently associated with shorter progression-free and overall survival. Conversely, the absence of PD-L1 expression at the end of treatment correlated with complete response, suggesting that PD-L1(+) CTCs may identify patients at higher risk of relapse who could benefit from adjuvant PD-1 inhibitor therapy.

In parallel, ctDNA dynamics under immune checkpoint blockade (ICB) provide complementary prognostic insights [97]. In recurrent or metastatic HNSCC, ctDNA clearance during treatment was strongly associated with improved disease control and survival, whereas early increases in ctDNA levels predicted progression, independently of PD-L1 status. These findings highlight the potential of integrating PD-L1(+) CTCs and ctDNA kinetics as synergistic, non-invasive biomarkers for prognostic refinement in the immunotherapy era of HNSCC.

Together, molecular and cellular components of the liquid biopsy offer a comprehensive framework for prognostic assessment in HNSCC, setting the stage for the discussion of ctDNA’s role in post-treatment surveillance and minimal residual disease monitoring, which is addressed in the following section.

#### 2.2.3. Prognostic Insights in HPV- and EBV-Related Head and Neck Cancers

The prognosis of patients with HPV-positive head and neck squamous cell carcinoma (HNSCC) differs substantially from that of HPV-negative disease, reflecting distinct biological behavior and treatment responsiveness. Circulating tumor DNA (ctDNA) has emerged as a promising biomarker capable of capturing these molecular differences and providing prognostic insights in this subgroup. Recent meta-analytic evidence [89] indicates that positive ctHPV DNA is associated with poorer overall survival but not with progression- or recurrence-free survival, underscoring its potential utility for prognostic stratification in HPV-driven tumors.

Consistent with these findings, data from the ARTSCAN III trial [98] demonstrated that ctHPV16-DNA kinetics during chemoradiotherapy represent an independent prognostic factor in HPV-related oropharyngeal cancer. Lower baseline ctHPV16-DNA levels and a more pronounced decline throughout treatment correlated with improved progression-free and overall survival, reflecting the prognostic value of both quantitative levels and clearance dynamics.

Complementary evidence supports the contribution of epigenetic alterations to prognostic stratification in HPV-positive oropharyngeal carcinoma. In a verification study of 252 HNSCC patients using quantitative methylation-specific PCR (Q-MSP) [71], three genes—*CALML5*, *DNAJC5G*, and *LY6D*—showed strong predictive ability for recurrence risk in HPV-associated disease. Methylation of these genes was detected in nearly all pre-treatment plasma samples and declined markedly after therapy, suggesting their potential as dynamic biomarkers for early recurrence detection and risk classification in HPV-driven OPSCC.

In parallel, analysis of circulating tumor cells (CTCs) showed that p16-positive CTCs were independently associated with reduced risk of disease progression and cancer-specific death, further supporting the role of viral-associated biomarkers in refining prognostic assessment [99]. Collectively, these findings highlight the potential of ctHPV16-DNA and p16-positive CTCs as complementary, non-invasive tools for risk stratification and treatment personalization in HPV-driven HNSCC.

Beyond HPV-related disease, similar mechanisms have been established in Epstein–Barr virus (EBV)-associated nasopharyngeal carcinoma (NPC), where circulating EBV DNA serves as a robust prognostic marker [89]. Elevated pre-treatment ctEBV DNA levels correlate with higher tumor burden and poorer survival, while post-treatment clearance is associated with favorable outcomes [100]. Large-scale studies in endemic regions have further demonstrated that integrating ctEBV DNA quantification into clinical staging systems enhances risk prediction and patient stratification compared with conventional TNM criteria. Together, evidence from both HPV- and EBV-driven HNSCC underscores the prognostic value of viral ctDNA as a dynamic biomarker that mirrors tumor biology, offering a foundation for more precise and individualized patient management.

### 2.3. Monitoring Tumor Dynamics and Therapeutic Response Through Circulating Biomarkers

Circulating biomarkers have emerged as versatile tools in the management of head and neck cancers. Their role extends beyond the detection of MRD to include dynamic assessment of treatment response [101,102,103,104,105]—whether to surgery, chemotherapy, radiotherapy or immunotherapy—and monitoring during follow-up [106,107,108] (Table 5). By providing real-time molecular insights, these biomarkers can complement conventional imaging, particularly in situations where radiological findings are uncertain, thus enabling earlier intervention and more tailored patient management. Indeed, in cases where after-treatment PET-CT showed limited accuracy, with inconclusive or ambiguous results, liquid biopsy and ctDNA detection represent excellent adjunct test for post-treatment PET to select patients at higher risk of recurrence [109], even in HPV-positive disease where ctHPVDNA has shown promising predictive values [110]. ctHPVDNA has proven more reliable than PET-CT following treatment in patients with OPSCC and its integration in post-treatment response assessment might lead to a decrease in unnecessary medical procedures, such as neck dissection [111]. In this context, the integration of circulating tumor HPV DNA with conventional imaging modalities further enhances post-treatment surveillance, as ctHPVDNA has been shown to detect recurrence earlier than PET-CT, improve the interpretation of equivocal radiologic findings and provide a high negative predictive value that may help avoid unnecessary invasive procedures.

#### 2.3.1. MRD and ctDNA in HNSCC (Table 6)

In Oncology, MRD refers to the presence of a small number of cancer cells remaining in the body after treatment, often undetectable by conventional clinical or radiological assessments, yet capable of regrowth and relapse [112]. Detection of MRD is crucial for prognosis and clinical decision-making, guiding early interventions and potentially improving survival. MRD can be assessed using DNA- or RNA-based molecular assays. DNA-based assays detect cancer-specific mutations in circulating tumor DNA (ctDNA) using next-generation sequencing (NGS). These assays can be tumor-informed, leveraging mutations from the patient’s own tumor, or tumor-agnostic, applying a fixed panel of frequently mutated cancer genes [113]. RNA-based assays detect fusion transcripts via reverse transcription polymerase chain reaction (RT-PCR) and are useful in specific contexts, such as large rearrangements, though they require careful RNA handling.

Recent studies highlighted ctDNA as a sensitive biomarker for detecting MRD in locally advanced head and neck squamous cell carcinoma (LA HNSCC). Both personalized and tumor-agnostic ctDNA assays can identify patients at high risk of recurrence, enabling earlier intervention. Personalized (tumor-informed) ctDNA assays have shown that post-treatment positivity occurs in 9.5% of patients and is associated with significantly worse survival, allowing recurrence to be identified approximately seven months before clinical detection [114]. Similarly, tumor-agnostic assays, which used a ctDNA panel targeting 26 frequently mutated SCCHN genes and HPV-16, were applied to 53 patients after curative treatment. In this cohort, post-treatment ctDNA positivity was observed in 41% of patients and correlated with substantially poorer outcomes, including a 2-year progression-free survival of 24% versus 87% in negative patients, as well as shorter median overall survival [115].

When dealing with surgically treated patients, ctDNA often precedes clinical or radiological relapse due to the detection of MRD. For example, in a cohort of patients with clinical recurrence, ctDNA detection preceded progression by 108–253 days [116]. In the SCANDARE study (NCT03017573) in 41 non-metastatic, resectable HNSCC patients undergoing curative surgery, ctDNA was detected in 51% at the time of surgery and 68% at recurrence. Longitudinal plasma analysis revealed mutations not found in tumor tissue in 57% of cases. Postoperative ctDNA positivity anticipated clinical recurrence by a median of 9.9 months in 63% of patients and was independently associated with recurrence when measured within 14 weeks after surgery (HR = 3.0, *p* = 0.03) [117]. In a subgroup analysis of the IMSTAR HN trial, liquid biopsy monitoring showed feasibility in resectable HNSCC, with cell-free DNA mutations after surgery identifying patients at risk for early relapse [118]. In HPV-positive oropharyngeal squamous cell carcinoma (OPSCC), ctHPVDNA has emerged as a highly sensitive noninvasive biomarker for MRD detection, treatment monitoring and recurrence prediction. Ferrier et al. demonstrated that ctDNA in blood and saliva dropped from 91% pre-treatment to 8% post-treatment, with persistent ctDNA strongly associated with residual tumor and recurrence [119]. Postoperative ctHPVDNA detection was associated with lower recurrence-free survival and correlated with adverse pathologic features, including lymphovascular invasion and extranodal extension [120]. Also, in a prospective study of 33 HPV-positive OPSCC patients, ctHPVDNA levels rapidly declined to <1 copy/mL by postoperative day 1 in low-risk patients, whereas elevated levels correlated with residual disease and adverse pathologic features, identifying recurrence up to two months before clinical diagnosis [121]. These studies collectively reinforce ctHPVDNA as a noninvasive MRD marker in HPV-positive HNSCC, with implications for early detection, recurrence prediction and treatment personalization. Amongst ongoing studies, the NeckTAR trial (NCT05710679) is a recruiting multicenter study aiming to predict MRD via ctDNA after potentiated radiotherapy for locally advanced HNSCC [122].

**Table 6 cancers-17-03974-t006:** Post-Treatment ctDNA and ctHPVDNA Biomarkers for MRD Detection and Recurrence Prediction in Head and Neck Cancer.

First Author, Year	Setting	Biomarker and Assay Type	N. Patients	Main Findings	Clinical Implications
**Honorè N., 2025** **[114]**	Post-treatment monitoring of LA HNSCC	Personalized, tumor-informed 16-plex PCR-NGS ctDNA assay	43 (50 enrolled)	Post-treatment ctDNA positivity (9.5%) predicted significantly worse RFS and OS	Post-treatment ctDNA positivity identifies patients at high risk of recurrence
**Marret G., 2025** **[117]**	Post-surgical and longitudinal surveillance in non-metastatic resectable HNSCC	Targeted NGS ctDNA sequencing on serial plasma samples (tumor and blood compared)	41	ctDNA detected in 51% at surgery and 68% at recurrence; MRD positivity within 14 weeks correlated with recurrence (HR = 3.0, *p* = 0.03); median lead-time to clinical relapse = 9.9 months.	ctDNA enables detection of MRD and ITH, offering a predictive tool for relapse and insights into tumor evolution beyond tissue sequencing.
**Ferrier ST, 2023** **[119]**	Post-treatment monitoring of HPV-positive HNC	ddPCR detection of HPV16/18/31/33/35/45 ctDNA in blood and saliva	60 HPV+ (17 HPV-controls)	ctDNA detection significantly higher pre-treatment (91%) vs. post-treatment (8%); strong blood–saliva concordance (93%). Persistent ctDNA associated with residual disease and recurrence.	Combined blood and saliva ctDNA testing is a non-invasive, sensitive biomarker for treatment response and early recurrence detection in HPV+ HNC
**Honorè N., 2023 [115]**	Post-treatment MRD detection in LA SCCHN	Tumor-agnostic 26-gene NGS plasma ctDNA assay (includes 2 HPV-16 genes)	53	MRD detected in 41% of patients after treatment. 2-year PFS: 23.5% (MRD+) vs. 86.6% (MRD–); median survival: 28.4 months (MRD+) vs. not reached (MRD–).	Tumor-agnostic ctDNA assay predicts progression and survival without tumor sequencing, enabling broad clinical implementation for MRD monitoring in LA SCCHN
**Flach S., 2022** **[116]**	Post-surgical surveillance in p16-negative HNSCC	Personalized RaDaR™ deep-sequencing ctDNA assay (tumour-specific variants from WES)	17	ctDNA detected in 100% of baseline samples and at very low VAF (0.0006%) post-surgery; ctDNA positivity preceded all clinical recurrences by 108–253 days	Personalized ctDNA assays are feasible and highly sensitive for MRD detection and early relapse prediction in surgically treated p16– HNSCC.
**Jonas H., 2022** **[118]**	Post-treatment and immunotherapy monitoring in localized HNSCC (IMSTAR-HN trial)	Tumor-specific digital droplet PCR (ddPCR) liquid biopsy based on NGS-identified mutations	19	Personalized ddPCR assays feasible in 17/18 patients; persistent or emerging ctDNA in ≥2 consecutive samples predicted relapse in most cases. Lead time to clinical recurrence up to 18 weeks. Patients achieving full ctDNA clearance had no relapse.	ddPCR-based ctDNA monitoring is a feasible and sensitive approach for identifying minimal residual disease and relapse risk in localized HNSCC
**Routman DM, 2022** **[120]**	Postoperative monitoring in HPV-positive OPSCC	Multianalyte PCR assay detecting ctHPVDNA	159 postop samples (32 paired pre/post)	Detectable postop ctHPVDNA strongly associated with worse RFS (*p* < 0.001) and correlated with adverse features (LVI, ENE)	Detectable postoperative ctHPVDNA before adjuvant therapy is a potential biomarker of residual disease and predictor of recurrence risk in HPV+ OPSCC.
**O’Boyle CJ, 2022** **[121]**	Post-surgical kinetics of HPV ctDNA in HPV-positive OPSCC	Custom ddPCR assay for HPV16/18/33/35/45 ctHPVDNA	33	ctHPVDNA levels cleared to <1 copy/mL by POD 1 in patients without residual disease; remained high (>350 copies/mL) in macroscopic residual disease; intermediate levels (1.2–58.4 copies/mL) corresponded to microscopic residual disease.	Postoperative day 1 ctHPVDNA levels reflect residual disease burden and may guide adjuvant therapy decisions in HPV+ OPSCC.
**Leung E., 2021** **[58]**	Retrospective analysis of prospective cohorts (cervical cancer + OPSCC)	HPV ctDNA analyzed by next-generation sequencing (HPV-seq) vs. dPCR	Multicenter cohorts (sample size not fully stated for OPSCC alone)	HPV-seq detects ctDNA at much lower levels than dPCR (<0.03 copies/mL); excellent correlation with dPCR (R^2^ = 0.95); 100% sensitivity for detecting recurrence post-treatment (specificity 67%); reliable genotyping in 100% of baseline samples; distinct fragmentomic signatures of ctDNA	Superior performance for low tumor burden and minimal residual disease (MRD) detection → strong potential for early relapse prediction and improved treatment response monitoring

**LA HNSCC,** Locally Advanced Head and Neck Squamous Cell Carcinoma; **PCR-NGS**, Polymerase Chain Reaction-Next Generation Sequency; **ctDNA**, Circulating Tumor DNA; **MRD**, Minimal Residual Disease; **ITH**, Intratumoral Heterogeneity; **HPV**, Human Papillomavirus; **HNC**, Head and Neck Cancer; **ddPCR**, Digital Droplets PCR; **OPSCC**; Oropharyngeal Squamous Cell Carcinoma; **RFS**, Recurrence-free Survival; **LVI**, Lymphovascular Invasion; **ENE**, Extranodal Extension.

#### 2.3.2. HPV-Positive OPSCC (Table 7)

Beyond its established role in detecting MRD, ctDNA also holds significant potential in post-treatment surveillance and follow-up, serving as a dynamic biomarker to predict therapeutic response and to identify patients at risk of early recurrence. This is particularly relevant in HPV-positive disease, where circulating viral DNA fragments can provide a highly sensitive and specific means to monitor treatment response and detect molecular recurrence earlier than imaging [123,124,125]. In a prospective cohort of 72 patients with HPV+/p16+ OPSCC, plasma cell-free HPV DNA (cfHPV-DNA) was evaluated as a biomarker for disease monitoring after curative treatment. CfHPV-DNA showed high baseline sensitivity (97.2%) and perfect concordance with tumor HPV genotypes. During follow-up, patients with persistently undetectable cfHPV-DNA experienced no recurrences, whereas cfHPV-DNA positivity anticipated radiologic or biopsy-confirmed relapse by up to 5 months [46]. In a real-world cohort of 399 OPSCC patients, plasma tumor tissue–modified viral (TTMV) HPV DNA testing showed high accuracy for diagnosis and surveillance of HPV-positive disease, with sensitivity of 91.5% and 88.4% and 100% specificity in both settings. The test detected recurrences a median of 47 days before clinical confirmation, supporting TTMV-HPV DNA as a reliable tool for early detection and monitoring of HPV-associated OPSCC [44]. Another study evaluated the clinical utility of TTMV HPV DNA testing during post-treatment surveillance, particularly in cases with indeterminate imaging or clinical findings. Among patients with equivocal results, TTMV-HPV DNA demonstrated high diagnostic accuracy (97.5%) in correctly determining recurrence status, enabling earlier confirmation of relapse compared with conventional assessments [126].

Dynamic changes in circulating HPV ctDNA during therapy may provide real-time insights into treatment efficacy. Building on this rationale, in 34 patients with stage III p16+ OPSCC undergoing chemoradiation, serial HPV ctDNA measurements were compared with MRI and FDG-PET imaging biomarkers. Low baseline ctDNA and an early rise at week 2 correlated with better PFS, while later ctDNA levels were not predictive, thus suggesting that early ctDNA kinetics may serve as a noninvasive biomarker of treatment response in HPV-positive OPSCC [127]. Consistent with these findings, a larger multi-institutional trial of 103 patients with HPV-associated OPSCC demonstrated that a rapid ctHPV-DNA clearance profile—defined by high baseline viral load and >95% clearance by day 28 of CRT—was strongly associated with disease control, whereas patients with slower clearance and adverse clinical risk factors had significantly higher rates of regional persistence or recurrence [128]. Together, these studies highlight the prognostic relevance of ctHPV-DNA kinetics during CRT and support its potential role in guiding treatment adaptation and de-intensification strategies for HPV-positive OPSCC. Early studies first demonstrated that plasma cfHPV-DNA levels reflect tumor burden and disease course in patients with HPV-driven OPSCC. In an observational cohort of 50 patients, cfHPV-DNA concentrations correlated with tumor size and declined after successful therapy, whereas persistent or rising levels were associated with residual or recurrent disease. These findings established the foundation for using cfHPV-DNA as a dynamic biomarker for treatment monitoring and early recurrence detection in HPV-positive OPSCC, later refined by more advanced ctHPV-DNA kinetic and fragmentomic approaches [100].

**Table 7 cancers-17-03974-t007:** Summary of Clinical Studies Investigating Circulating HPV DNA as a Biomarker in HPV-Associated Oropharyngeal Squamous Cell Carcinoma (OPSCC).

First Author, Year	Setting	Biomarker and Assay Type	N. Patients	Main Findings	Clinical Implications
**Oldaeus Almeren A., 2025** **[125]**	Multicenter, prospective cohort of HPV+ OPSCC/HNCUP undergoing definitive (chemo)radiotherapy	HPV genotype-specific ddPCR ctHPV-DNA in plasma	51	Baseline ctDNA correlated with total tumour and nodal volume. Undetectable ctHPV-DNA at follow-up corresponded to favorable radiologic response.	ctHPV-DNA is a promising biomarker for treatment evaluation, correlates with tumor burden and could complement radiologic assessment in HPV+ OPSCC/HNCUP.
**Roof SA, 2024** **[126]**	Retrospective multicenter cohort, HPV-associated OPSCC, post-treatment surveillance	Plasma TTMV-HPV DNA assay	543	210/543 patients had clinically indeterminate findings (CIFs). TTMV-HPV DNA testing accurately resolved 97.5% (77/79) of indeterminate cases; minimal discordance with clinical outcomes (0.6%).	TTMV-HPV DNA is a highly accurate tool for resolving indeterminate findings during HPV+ OPSCC surveillance, improving clinical decision-making and potentially reducing overtreatment.
**Ferrandino RM, 2023** **[44]**	Retrospective observational cohort, HPV-associated OPSCC	Plasma TTMV-HPV DNA	399 (163 diagnostic, 290 surveillance)	Diagnostic sensitivity 91.5%, specificity 100%; surveillance sensitivity 88.4%, specificity 100%; median lead time to pathologic confirmation = 47 days.	TTMV-HPV DNA is a highly specific and sensitive biomarker for both diagnosis and surveillance of HPV-associated OPSCC.
**Jakobsen KK, 2023** **[46]**	Prospective cohort, HPV+/p16+ OPSCC, serial follow-up	Droplet digital PCR (ddPCR) targeting 8 HPV genotypes in plasma cfHPV-DNA	72 (54 with serial follow-up)	Baseline cfHPV-DNA sensitivity 97.2%, copy number correlated with tumor stage, 100% genotype concordance with tumor. cfHPV-DNA detected 97–166 days before recurrence. Patients with undetectable cfHPV-DNA did not recur.	Serial plasma cfHPV-DNA measurement is a highly sensitive, clinically applicable surveillance tool for early detection of recurrence in HPV+/p16+ OPSCC.
**Cao Y.,2022** **[127]**	Randomized trial, AJCC8 stage III p16+ OPSCC, during chemoradiation	Serial plasma HPV ctDNA, correlated with MRI (DCE, DWI) and FDG-PET imaging biomarkers	34	Low pretreatment ctDNA and early increase at week 2 associated with superior FFP. ctDNA at weeks 4–7 not predictive. ctDNA correlated with tumor subvolumes on MRI and PET	Early ctDNA kinetics during chemoradiation can serve as a predictive biomarker of therapy response, potentially complementing imaging in stage III p16+ OPSCC.
**Akashi K, 2021** **[124]**	p16-positive HPV-associated oropharyngeal cancer, multiple timepoints during treatment	Digital PCR targeting HPV-derived ctDNA in plasma	25	HPV ctDNA detected in 14/25 patients (56%); all became ctDNA-negative after initial treatment. ctDNA detected at time of recurrence in 2 patients.	HPV-derived ctDNA is a minimally invasive biomarker for monitoring treatment response and predicting recurrence in p16-positive OPSCC.
**Chera BS,2020** **[128]**	Multi-institutional prospective biomarker trial, p16+ HPV-associated OPSCC, definitive chemoradiotherapy	Serial plasma HPV ctDNA (types 16/18/31/33/35) measured by optimized multianalyte ddPCR	103	Defined a favorable clearance profile (>200 copies/mL baseline and >95% clearance by day 28) associated with 0% recurrence; unfavorable profile + adverse clinical factors had 35% recurrence.	Rapid ctHPVDNA clearance predicts disease control after CRT; may guide de-intensified therapy selection in HPV+ OPSCC.
**Reder H., 2020** **[100]**	Observational study, OPSCC patients	Plasma cfHPV-DNA targeting E6/E7 oncogenes, quantified by real-time qPCR	50	cfHPV-DNA correlated with tumor size. Post-treatment decline in patients without recurrence; persistent/increased cfHPV-DNA associated with residual disease or relapse.	Plasma cfHPV-DNA can monitor therapy response and detect minimal residual disease

**HPV**, Human Papillomavirus; **OPSCC**, Oropharyngeal Squamous Cell Carcinoma; **HNCUP**, Head and Neck Cancer of Unknown Primary; **ddPCR**, Droplet Digital Polymerase Chain Reaction; **ctDNA**, Circulating Tumor DNA; **cfHPV-DNA**, Cell-free HPV DNA; **TTMV-HPV DNA**, Tumor Tissue Modified Viral HPV DNA; **FFP**, Freedom From Progression; **qPCR**, Quantitative Polymerase Chain Reaction; **AJCC8**, American Joint Committee on Cancer, 8th edition; **DCE**, Dynamic Contrast-Enhanced; **MRI**, Magnetic Resonance Imaging; **DWI**, Diffusion-Weighted Imaging; **FDG-PET**, Fluorodeoxyglucose Positron Emission Tomography; **CIFs**, Clinically Indeterminate Findings; **CRT**, Chemoradiotherapy.

#### 2.3.3. Non-Oropharyngeal HNSCC

For what concerns non-oropharyngeal sites of disease, in Japanese patients with oral SCC (OSCC) total cell-free DNA (cfDNA) was analyzed preoperatively, postoperatively and during follow-up. Patients with stable disease showed minimal changes in cfDNA levels, whereas those who developed systemic metastases exhibited rapid increases [129]. Always in the context of OSCC, salivary microRNAs are emerging noninvasive biomarkers, complementing cfDNA analysis to monitor disease and predict recurrence, though standardized methods are needed for routine clinical use [130].

Circulating cell-free Epstein–Barr virus (cfEBV) DNA has emerged as a highly sensitive and specific biomarker for detecting residual disease and early recurrence in nasopharyngeal carcinoma (NPC), offering the potential to guide more targeted follow-up. Integrating cfEBV DNA into surveillance strategies may improve the efficiency and cost-effectiveness of post-treatment monitoring, reducing unnecessary imaging while enabling timely detection of recurrence [131,132,133,134]. Moreover, several studies have demonstrated the prognostic value of cfEBV DNA dynamics: post-induction chemotherapy plasma EBV DNA levels predict outcomes in locoregionally advanced NPC [135]; pre- and post-treatment EBV DNA levels are strongly correlated with prognosis [136]; dynamic changes in plasma EBV DNA, alongside peripheral blood immune parameters, identify distinct risks of treatment failure [137] and serial post-IMRT undetectable plasma EBV DNA provides robust prognostic information for long-term outcomes [138]. Collectively, these findings underscore the role of cfEBV DNA not only as a valuable biomarker for early detection but also for post-treatment prognostication in NPC, supporting its integration into personalized surveillance protocols. Although most evidence on cfEBV DNA originates from endemic regions, emerging studies from non-endemic areas have similarly demonstrated its clinical utility. In these low-incidence settings, cfEBV DNA has shown value for MRD detection, treatment response monitoring and early recurrence identification, supporting its applicability beyond endemic populations. These findings broaden the global relevance of cfEBV DNA and highlight its potential integration into surveillance workflows regardless of regional NPC prevalence.

#### 2.3.4. Induction Therapy and Recurrent/Metastatic (R/M) Settings

Furthermore, ctDNA offers the opportunity to dynamically monitor tumor response during active treatment, including neoadjuvant or induction chemotherapy, allowing early identification of patients who are responding or not to therapy. Huttinger et al. [139] conducted a pilot study of 17 HPV-positive OPSCC patients treated with induction chemotherapy (IC) followed by chemoradiation (CRT), where HPV ctDNA levels were measured alongside PET-CT. In the results HPV ctDNA clearance outperformed imaging in assessing treatment response. In the context of neo-adjuvant immunotherapy as well, liquid biopsies, especially when combined with nanotechnology, have shown the potential for highly sensitive prediction [140]. Furthermore, in 50 HNSCC patients treated with neoadjuvant PD-1 therapy plus chemotherapy, blood immune profiling and plasma cytokines distinguished responders from non-responders. Responders had more CD103^−^CD8^+^ central memory T cells and higher IL-5/IL-13, while non-responders had more Temra cells and elevated CCL3, CCL4 and MMP7. Therapy activated both subsets and a predictive model combining these markers achieved high accuracy (AUC = 0.92), demonstrating that liquid biopsy can noninvasively guide early response and personalized immunotherapy [141].

The role of ctDNA has also been reported in the setting of recurrent/metastatic (R/M) disease [142]. In a study of 16 patients with R/M HNSCC receiving immunotherapy, ctDNA negativity during treatment strongly predicted disease control, longer progression-free survival and overall survival, highlighting ctDNA as a potent noninvasive biomarker for monitoring response [97]. Haring et al. [143] showed that in patients with R/M HNSCC receiving immunotherapy, serial ctDNA monitoring showed that ctDNA negativity predicts better disease control, progression-free survival and overall survival, making it a valuable noninvasive biomarker for treatment response.

In summary, ctDNA represents a dynamic and versatile biomarker for monitoring treatment response in HNSCC. Serial measurements provide real-time insights into tumor behavior, allowing early identification of responders and non-responders across different therapeutic settings, including surgery, chemoradiation, immunotherapy and induction or neoadjuvant treatments. Post-treatment ctDNA positivity is strongly associated with minimal residual disease and higher risk of recurrence, whereas rapid clearance correlates with favorable outcomes, particularly in HPV-positive OPSCC. By complementing conventional imaging, ctDNA enables timely, noninvasive assessment of therapeutic efficacy and offers significant potential to guide personalized treatment strategies and optimize patient management.

### 2.4. Guiding Precision Medicine Using Circulating Biomarkers in Head and Neck Cancer

Integrating liquid biopsy with targeted therapy and precision medicine represents an innovative approach in Oncology, aiming to tailor treatments based on the molecular characteristics of each patient’s disease. ctDNA has emerged as a powerful tool to identify patients eligible for targeted therapies by revealing actionable genetic alterations in a minimally invasive manner. Serial ctDNA analysis enables dynamic monitoring of tumor evolution, allowing early detection of emerging mechanisms of treatment resistance. Whole exome sequencing (WES) of cfDNA has successfully identified actionable alterations in the majority of patients, particularly when tissue biopsies are inaccessible, providing a rapid and practical approach to guide precision oncology [144].

In oral cancer, often diagnosed at advanced stages and associated with poor prognosis due to recurrence, metastasis and treatment resistance, exosomes—small endosome-derived lipid nanoparticles—reflect the molecular profile of their donor cells and serve as liquid biopsy tools for early diagnosis. Beyond their diagnostic potential, exosomes actively contribute to tumor progression by transferring bioactive cargoes to recipient cells and have been implicated in therapy resistance, including drug efflux, highlighting their dual role as predictive biomarkers, therapeutic targets and delivery vectors [145].

In HNSCC retrospective studies revealed that ctDNA profiling identified actionable alterations in approximately two-thirds of patients, with specific changes in DNA repair genes and TP53 associated with poor overall survival. Notably, concordance between ctDNA and tumor tissue DNA (tDNA) was limited, emphasizing the unique clinical value of ctDNA as a real-time, non-invasive biomarker [67]. ctDNA has also proven useful in guiding post-treatment decisions. Post-treatment HPV cfDNA accurately identified OPSCC patients with residual disease after non-surgical therapy, guiding salvage neck dissection [146]. In HPV-positive OPSCC, post-surgical ctHPV-DNA testing demonstrated high specificity and negative predictive value, effectively identifying patients unlikely to require adjuvant (chemo)radiation, thereby supporting individualized treatment strategies [147]. Furthermore, secondary analyses of the DART Phase 3 trial indicated that postoperative ctHPV-DNA can guide adjuvant treatment de-escalation, enhancing personalized therapy [148].

Collectively, these findings highlight the potential of liquid biopsy—including ctDNA, cfDNA and exosomes—to guide precision oncology and targeted therapy in HNSCC and oral cancer.

Within the broader framework of using ctDNA for diagnosis, risk stratification, treatment response assessment and precision oncology, it is important to highlight the distinct biomarker landscape of HPV-negative HNSCC. This subgroup, characterized by a poorer prognosis and a more complex mutational profile compared with HPV-positive tumors, lacks viral biomarkers such as ctHPV-DNA, making tumor-derived genomic and epigenomic alterations particularly relevant. Among these, TP53 mutations—the most common genetic alteration in HPV-negative HNSCC—represent a promising circulating biomarker. Detection of TP53-mutated ctDNA reflects tumor burden and clonal evolution and several studies have shown that its persistence after curative-intent treatment is associated with minimal residual disease and early relapse. Despite these encouraging observations, the use of TP53-mutated ctDNA for routine clinical decision-making remains investigational and is not yet incorporated into standard management pathways.

Similarly, ctDNA methylation markers, such as promoter hypermethylation of genes including CDKN2A/p16, MGMT and DAPK1, have shown potential for early detection, prognostic stratification and monitoring of disease progression in HPV-negative patients. Aberrant methylation signatures can be reliably detected in circulating DNA and may complement mutational analyses by capturing early epigenetic dysregulation. However, these assays remain largely confined to research settings, with substantial work still needed to standardize workflows, validate targets and assess their incremental clinical utility.

In summary, while TP53 mutations and methylation-based ctDNA biomarkers hold significant promise for improving the diagnostic and prognostic evaluation of HPV-negative HNSCC, their integration into routine clinical practice is still premature. At present, patient management continues to rely primarily on clinical assessment, imaging and tissue biopsy, with circulating genomic and epigenomic markers representing emerging tools that require further prospective validation before widespread adoption.

## 3. Current Challenges and Future Directions

Despite the recent successes of cancer therapy, numerous obstacles remain. Cancers vary greatly not only at patient, tissue and cellular levels but also at molecular level. Because of this multiscale heterogeneity, effective treatments that not only differentiate between cancerous and healthy tissues but also target a wide variety of tumor subclones are difficult to develop. Most treatments either take advantage of a specific biological characteristic shared by cancer cells (e.g., their propensity for rapid division) or indiscriminately eradicate every cell in an area of interest. This multiscale complexity underscores the need for real-time, minimally invasive tools to monitor tumor dynamics, such as circulating biomarkers including ctDNA, CTCs, and EVs, which can inform treatment decisions, detect minimal residual disease and anticipate resistance [149].

The performance of liquid biopsy is highly influenced by pre-analytical variables, including the type of blood collection tubes, processing time, and temperature control, all of which impact ctDNA stability and plasma quality [150]. Using cfDNA-stabilizing tubes reduces leukocyte contamination and increases DNA yield compared with standard EDTA tubes [151]. Rapid processing, ideally within six hours, is recommended to preserve molecular integrity [152].

Analytical variability further affects assay reliability. Digital PCR (dPCR) improves detection of low-frequency variants compared to qPCR but may miss complex genomic alteration [153]. Next-generation sequencing (NGS) allows comprehensive genomic profiling, capturing a wide range of mutations, yet requires sophisticated bioinformatics to interpret samples with low tumor fractions [154]. Sample preservation methods, including stabilizing reagents and microfluidic storage systems, help maintain assay consistency [155,156], while standardized workflows have improved reproducibility across laboratories [157].

Despite these technical advances, several challenges persist. Inter-laboratory variability remains a major barrier to data harmonization [158], and the absence of universal reference standards limits robust assay validation [159]. Differences in workflow and sequencing platforms contribute to heterogeneous results [160]. Moreover, although the cost of liquid biopsy testing remains high, its potential for earlier recurrence detection may ultimately improve cost-effectiveness and patient outcomes [161].

Tissue biopsy continues to be the gold standard for HNSCC diagnosis, staging, and grading, providing critical histopathological information and insights into tumor heterogeneity [162]. However, tissue sampling is invasive, often difficult to repeat, and cannot capture dynamic molecular changes or real-time treatment responses [163].

Looking ahead, the integration of tissue and liquid biopsy into complementary clinical workflows holds significant promise. Combining comprehensive histopathological assessment with real-time molecular surveillance could enhance precision oncology in HNSCC, enabling earlier intervention, more personalized treatment, and improved long-term outcomes. Current research is focused on optimizing assay performance, refining clinical implementation, and advancing standardization, laying the groundwork for a future in which liquid biopsy becomes a routine, actionable tool in patient management.

Ongoing initiatives, such as the Blood Profiling Atlas in Cancer (BLOODPAC) Consortium and other collaborative efforts, are working to address the significant lack of standardization in liquid biopsy workflows. These groups are developing consensus protocols for pre-analytical handling, including standardized procedures for blood collection, stabilization, processing times and storage conditions. Furthermore, these initiatives promote standardized analytical techniques, the use of reference materials and common bioinformatics pipelines to ensure ctDNA measurements are consistent across different laboratories [164,165,166]. Such efforts are essential for improving the reproducibility of liquid biopsy results, enabling robust assay validation, and accelerating the integration of these tests into routine clinical practice, particularly for HNSCC. By adopting these standardized approaches, future studies can more reliably compare results, optimize assay performance and ultimately improve patient management through precise and timely molecular monitoring.

### Limitations

This narrative review has several inherent limitations. First, as a non-systematic review, it does not employ a formalized methodology for study selection, which may introduce selection bias and limit reproducibility. Second, the included studies vary widely in terms of patient populations, assay platforms and clinical settings, which may contribute to heterogeneity and limit the generalizability of the conclusions. Moreover, many studies are limited by small sample sizes, single-center designs and retrospective analyses, which should be considered when interpreting the results. Third, publication bias cannot be excluded, as studies with significant or positive findings are more likely to be published and included. Finally, while efforts were made to provide a comprehensive overview, some emerging biomarkers or novel technologies may not be fully represented, reflecting the rapidly evolving nature of liquid biopsy research in head and neck cancer.

Although various circulating biomarkers can theoretically be detected through liquid biopsy, the current evidence in head and neck cancer is predominantly centered on ctDNA and viral ctDNA. As a result, most of the studies included in our review and summarized in the tables focus on these two biomarker classes. In contrast, data regarding other circulating biomarkers—such as CTCs, cfRNA, miRNAs and extracellular vesicles—remain more limited, heterogeneous and at an earlier stage of clinical validation, which restricts their representation and comparative analysis. Overall, while multiple circulating biomarkers are under investigation, their clinical utility varies considerably due to intrinsic biological differences and variable assay maturity. ctDNA represents the most clinically advanced biomarker, offering high specificity and the ability to capture tumor-derived genetic alterations, making it particularly valuable for treatment response monitoring, MRD detection and recurrence prediction, although sensitivity may be reduced in tumors with low shedding, including some HPV-negative cases. CTCs provide direct insight into tumor dissemination and metastatic potential, yet their rarity and the lack of standardized enrichment platforms limit their usability. EVs and RNA-based biomarkers (cfRNA and miRNAs) show promise due to their potential to reflect transcriptional activity and intercellular signaling, but the evidence remains heterogeneous and analytical workflows are not yet standardized. Interestingly, a research article by Zhang et al. [167] reviewed the role of non-coding RNAs (ncRNAs) in mediating drug resistance in head and neck cancer and suggested their potential as biomarkers and therapeutic targets.

Collectively, ctDNA—particularly viral ctDNA in HPV-positive disease—remains the most robust and clinically validated circulating biomarker in HNSCC, whereas CTCs, EVs and RNA-derived markers represent emerging but still investigational tools requiring further methodological refinement and prospective validation. Preclinical studies using patient-derived xenograft (PDX) models have supported these findings [168], as they preserve tumor heterogeneity and faithfully recapitulate treatment responses, providing a platform to investigate biomarker dynamics, resistance mechanisms and novel therapeutic strategies. Such models complement clinical observations, offering mechanistic insights and strengthening the translational relevance of circulating biomarker research in HNSCC.

It is important to acknowledge that in recent years several reviews have addressed the role of circulating DNA in head and neck cancers. However, unlike many of these works, our review encompasses all head and neck cancer sites across different clinical settings, including recurrent and metastatic disease. In addition, rather than discussing diagnosis, prognosis and treatment response as isolated aspects, we provide an integrated workflow illustrating how ctDNA can be incorporated throughout the entire diagnostic–therapeutic pathway. Furthermore, an additional strength of our review is that, unlike most existing articles that examine only selected circulating biomarkers or restrict their analysis to specific clinical applications, we offer a comprehensive and comparative overview of all major circulating biomarker classes and related technologies, thereby providing a broader translational perspective on how liquid biopsy can be implemented across the full spectrum of HNSCC management.

## 4. Conclusions

Liquid biopsy represents a promising and minimally invasive approach for the detection, monitoring and management of head and neck cancers. Circulating biomarkers, including ctDNA, ctHPV-DNA, CTCs, cfRNA and extracellular vesicles, provide dynamic insights into tumor burden, treatment response and minimal residual disease, complementing conventional tissue biopsy and imaging. Despite technical challenges and the need for further standardization, integrating liquid biopsy into clinical workflows has the potential to enable earlier intervention, guide personalized therapy and improve patient outcomes. Ongoing research and clinical validation will be critical to fully realize its role in precision oncology for head and neck cancer.

## Figures and Tables

**Figure 1 cancers-17-03974-f001:**
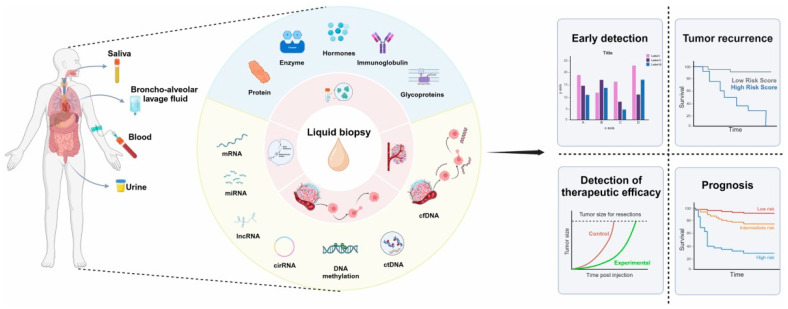
Clinical application of biomarkers in Head and Neck Squamous Cell Carcinoma (HNSCC). Reproduced from Ma L. et al. *Liquid biopsy in cancer: current status, challenges and future prospects*. Sig Transduct Target Ther **9**, 336 (2024) [38]. https://doi.org/10.1038/s41392-024-02021-w, under the Creative Commons Attribution 4.0 International License (CC BY 4.0). https://creativecommons.org/licenses/by/4.0/.

**Table 1 cancers-17-03974-t001:** Comparison of major ctDNA detection technologies based on diagnostic performance, cost, clinical application and feasibility.

Method	Sensitivity	Specificity	Cost	Turnaround Time	Clinical Use	Economic Feasibility
qPCR	75–85%	>90%	Low	24–48 h	Targeted testing, initial screening	High
ddPCR	>95%	>98%	Medium	24–72 h	Real-time disease monitoring and MRD detection	Cost-effective for routine use
NGS (hybrid-capture)	>99%	~97%	Very high	7–10 days	Comprehensive genomic profiling	Expensive; requires specialized support
NGS (amplicon-based)	~95%	~96%	High	5–7 days	Detection of actionable mutations and resistance	Feasible in specialized centers

**Table 5 cancers-17-03974-t005:** Circulating Biomarkers for Monitoring Treatment Response and Recurrence in Head and Neck Squamous Cell Carcinoma.

First Author, Year	Setting	Biomarker and Assay Type	N. Patients	Main Findings	Clinical Implications
**Glenn JH, 2024** **[107]**	Predominantly newly diagnosed HPV-negative locally advanced HNSCC	ctDNA, tumor-informed 16-plex PCR (Signatera, Natera)	100	Posttreatment ctDNA positivity strongly correlated with worse progression-free survival (HR 7.33)	Tumor-informed ctDNA is feasible in HPV-negative HNSCC; posttreatment ctDNA positivity identifies patients at high risk of progression and could guide therapy decisions
**Chikuie N., 2022 [101]**	Post-treatment follow-up of HNSCC after radical therapy	ctDNA from serial plasma samples using NGS	20	ctDNA detected in 5/7 patients with recurrence but not in 13 recurrence-free cases. Post-treatment ctDNA positivity associated with shorter RFS (9.6 ± 9.1 vs. 20.6 ± 7.7 months, *p* < 0.01).	ctDNA monitoring after radical treatment can identify early recurrence and predict poor prognosis earlier than radiologic methods
**Zhang X., 2022** **[103]**	Treatment-naïve head and neck cancer	CTCs isolated using a spiral microfluidic device and characterized by immunofluorescence staining	119	Higher CTC counts correlated with advanced stage and incomplete treatment response	Baseline CTC count serve as independent predictors of treatment response and prognosis
**Kogo R., 2022** **[104]**	HNSCC patients post-curative treatment	Individualized ctDNA analysis using tumor-informed SCC panel and monitored longitudinally via dPCR	26	Longitudinal ctDNA positivity after treatment predicted relapse in all 7 patients; ctDNA negativity predicted no recurrence in 11 patients. Significant difference in prognosis between positive vs. negative ctDNA (*p* < 0.0001).	Individualized ctDNA monitoring via dPCR can serve as a sensitive biomarker for early detection of relapse and treatment response
**Post CM, 2021** **[111]**	Post-treatment in HPV-positive OPSCC	ctHPVDNA, commercially available blood test	25	ctHPVDNA higher specificity (96%) and lower false positive rate (4%) than PET/CT (56% specificity, 44% false positives) at 3–6 months after treatment.	ctHPVDNA is a more reliable biomarker than PET/CT for post-treatment monitoring in HPV+ OPSCC
**Hilke FJ, 2020** **[102]**	Locally advanced HNSCC treated with definitive radiochemotherapy	ctDNA assessed via deep sequencing with UMI-based error suppression	20	ctDNA levels correlated with gross tumor volume (*p* = 0.032) and decreased progressively with treatment. Persistent ctDNA at first follow-up predicted later recurrence. Circulating HPV DNA showed similar kinetics, disappearing after therapy.	ctDNA serves as dynamic, minimally invasive biomarkers for real-time monitoring of treatment response during RCT
**Verma T, 2020** **[105]**	Locally advanced HNSCC monitored during chemoradiation	cfDNA quantified by β-globin real-time PCR	24 HNSCC patients, 16 healthy controls	cfDNA significantly elevated in HNSCC vs. controls. In responders, cfDNA decreased; in non-responders, cfDNA increased during 3-month follow-up.	cfDNA may serve as a non-invasive biomarker for monitoring response to chemoradiotherapy and potentially predicting early treatment outcomes in HNSCC
**Egyud M, 2019** **[108]**	Post-curative therapy follow-up	ctDNA, patient-specific mutations identified via tumor sequencing and tracked in plasma	8	Baseline ctDNA detected in 6/8 patients; recurrence occurred in 4 patients, 2 of whom had ctDNA detected prior to clinical recurrence	ctDNA may allow early detection of recurrence, improved prognostication, and guide modification of treatment strategies after curative therapy

**HPV,** Human Papillomavirus; **HNSCC**; Head and Neck Squamous Cell Carcinoma; **ctDNA**, Circulating tumor DNA; **PCR**, Polymerase Chain Reaction; **NGS**, Next Generation Sequencing; **RFS**, Recurrence-free Survival; **CTC,** Circulating Tumor Cells; **dPCR**, digital Polymerase Chain Reaction; **RCT**, Radiochemotherapy; **cfDNA**, Circulating Free DNA.

## Data Availability

No new data were created or analyzed in this study. Data sharing is not applicable to this article.

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
