# Peer review of "Liquid Biopsy and Circulating Biomarkers in Head and Neck Cancer: Advancing Non-Invasive Detection and Tailored Management"

_cancers, 2025, doi:10.3390/cancers17243974_

Round 1
Reviewer 1 Report
Comments and Suggestions for Authors
The present review article aims to provide an overview of the adoption of liquid biopsy and circulating biomarkers in non-invasive detection and management of head and neck squamous cell carcinoma (HNSCC).
The writing of the article and the structuring of the content are acceptable. Several tables provide an overview of studies conducted in HNSCC.
Below are the comments I made on the manuscript.
- The enumeration of subtitles can be helpful for a better follow-up of the content.
- Please use a unique abbreviation of head and neck squamous cell carcinoma, as both HNSCC and SCCHN are used in the text. The same applies to cfDNA/ccfDNA.
- Since there are many similar, recently published review articles on the role of circulating biomarkers in HNSCC, the authors should discuss the differences between their article and others.
- On page 3, the authors state, “Various biomarkers (e.g., cell-free DNA, circulating tumor DNA, circulating tumor cells, cell-free RNA, miRNAs, exosomes, and viral circulating tumor DNA) can be detected in head and neck tumors through liquid biopsy, offering valuable insights into tumor burden, molecular alterations, treatment response and disease recurrence”. However, most of the studies cited in the manuscript and presented in the tables focus primarily on circulating tumor DNA and viral circulating tumor DNA. This point should be clarified.
- ctDNA analysis in relation to therapy response in HNSCC should be described in more detail.
Author Response
We would like to sincerely thank the reviewer for providing such valuable comments. They have allowed us to develop a more comprehensive and thorough version of the manuscript.
The enumeration of subtitles can be helpful for a better follow-up of the content.
R: Subtitles have been enumerated, as you suggested.
Please use a unique abbreviation of head and neck squamous cell carcinoma, as both HNSCC and SCCHN are used in the text. The same applies to cfDNA/ccfDNA
R: Unique abbreviations have been adopted throughout the text for both Head and Neck Squamous Cell Carcinoma (HNSCC) and circulating free DNA (cfDNA), following your suggestion.
Since there are many similar, recently published review articles on the role of circulating biomarkers in HNSCC, the authors should discuss the differences between their article and others.
R: Based on your comment, we have provided a proper paragraph in the 3rd section that clearly outlines how our review differs from other published articles on circulating biomarkers in HNSCC and highlights the specific contributions of our work.
“It is important to acknowledge that in recent years several reviews have addressed the role of circulating DNA in head and neck cancers. However, unlike many of these works, our review encompasses all head and neck cancer sites across different clinical settings, including recurrent and metastatic disease. In addition, rather than discussing diagnosis, prognosis and treatment response as isolated aspects, we provide an integrated workflow illustrating how ctDNA can be incorporated throughout the entire diagnostic–therapeutic pathway. Furthermore, an additional strength of our review is that, unlike most existing articles that examine only selected circulating biomarkers or restrict their analysis to specific clinical applications, we offer a comprehensive and comparative overview of all major circulating biomarker classes and related technologies, thereby providing a broader translational perspective on how liquid biopsy can be implemented across the full spectrum of HNSCC management”.
On page 3, the authors state, “Various biomarkers (e.g., cell-free DNA, circulating tumor DNA, circulating tumor cells, cell-free RNA, miRNAs, exosomes, and viral circulating tumor DNA) can be detected in head and neck tumors through liquid biopsy, offering valuable insights into tumor burden, molecular alterations, treatment response and disease recurrence”. However, most of the studies cited in the manuscript and presented in the tables focus primarily on circulating tumor DNA and viral circulating tumor DNA. This point should be clarified.
R: Thank you for this valuable observation. We agree that most of the studies included in our tables and cited throughout the manuscript primarily focus on circulating tumor DNA and viral circulating tumor DNA. This reflects the current state of the literature, as ctDNA and viral ctDNA remain the most extensively investigated and clinically advanced circulating biomarkers in HNSCC. To address this point, we have clarified this aspect in the revised manuscript in the Limitations section by explicitly stating that, although several circulating biomarkers can theoretically be detected through liquid biopsy, the available evidence is predominantly centered on ctDNA and viral ctDNA. We have also added a brief explanation that data on other biomarker classes (including CTCs, cfRNA, miRNAs and EVs) remain more limited and heterogeneous, which explains their lower representation in the tables.
ctDNA analysis in relation to therapy response in HNSCC should be described in more detail.
R: We thank the reviewer for this valuable suggestion. In the revised manuscript, we have expanded section 2.3 to provide a more detailed discussion of ctDNA dynamics in relation to therapy response in HNSCC. We now describe its role across multiple clinical settings, including post-treatment monitoring and minimal residual disease detection, HPV-positive OPSCC, non-oropharyngeal HNSCC and induction, neoadjuvant, or recurrent/metastatic treatments. A new concluding paragraph has been added to summarize the dynamic and predictive value of ctDNA: serial measurements of ctDNA provide real-time insights into tumor behavior, allow early identification of responders and non-responders and complement conventional imaging. Post-treatment ctDNA positivity correlates with higher risk of recurrence, while rapid clearance is associated with favorable outcomes, particularly in HPV-positive OPSCC. Overall, ctDNA enables timely, noninvasive assessment of therapeutic efficacy and supports personalized treatment adaptation.
Reviewer 2 Report
Comments and Suggestions for Authors
refer to attachment

Author Response
Major comments
The review provides a comprehensive and well-structured overview of the current evidence on liquid biopsy in HNSCC, covering a wide range of biomarkers and clinical applications. As a narrative review, its methodology for study selection is not explicit, which introduces potential for selection bias and limits reproducibility. Adopting a systematic review approach (or at minimum, detailing search terms, databases, and inclusion/exclusion criteria) would significantly enhance its rigor and scholarly impact.
R: We thank the reviewer for highlighting this important point. We acknowledge that our review is a narrative review and as such, does not follow a fully systematic methodology. However, to ensure transparency and reproducibility, we have now explicitly described our literature search strategy in the paper. We conducted a comprehensive search of PubMed, Embase and Scopus databases using the following search string: (ctDNA OR "circulating tumor DNA" OR "liquid biopsy") AND ("head and neck cancer" OR "head and neck neoplasm" OR "head and neck tumor" OR HNSCC OR "head and neck squamous cell carcinoma"). Inclusion criteria were English-language articles published up to 2025, studies reporting on circulating biomarkers in HNSCC and studies addressing diagnosis, prognosis or therapeutic monitoring. Exclusion criteria were non-English publications, conference abstracts without full text, studies not related to HNSCC or studies focusing exclusively on animal models or in vitro experiments. While the review remains narrative in nature, these steps ensure that the selection process is transparent and reproducible, minimizing potential selection bias.
The manuscript offers an excellent, in-depth analysis of the robust data for ctHPV-DNA, rightly highlighting it as a highly sensitive application of liquid biopsy. However, this focus can overshadow the distinct challenges and current evidence gaps for HPV-negative HNSCC, which often has a poorer prognosis and a more complex mutational landscape. The review would be more balanced by explicitly comparing and contrasting the utility of different biomarkers (e.g., TP53 mutations, methylation) in the HPV-negative population.
R: We thank the reviewer for the suggestion; we have provided a proper paragraph before the conclusive section.
The manuscript correctly identified the lack of standardization as a major challenge limiting clinical implementation, mentioning pre-analytical variables and inter-laboratory variability. This critical point deserves a more thorough discussion. The authors could elaborate on specific ongoing efforts (e.g., initiatives by the Blood Profiling Atlas in Cancer or other consortia) to establish standardized protocols for sample collection, processing, and analysis to guide future research and clinical adoption.
R: We thank the reviewer for this important suggestion. We agree that standardization is a critical factor for the clinical implementation of liquid biopsy and that ongoing efforts by consortia and initiatives warrant discussion. In the revised manuscript, we have expanded the section on current challenges to include specific examples of initiatives aimed at establishing standardized protocols for sample collection, processing and analysis, thereby guiding future research and facilitating clinical adoption.
The review successfully synthesized findings from a large number of studies, effectively using tables to summarize key results and clinical implications. However, the quality and limitations of the cited studies (e.g., small sample sizes, single-center design, retrospective nature) are not critically discussed.
R: We thank the reviewer for this comment. We agree that the quality and design of the included studies represent an important consideration. In the revised manuscript, we have clarified in the Limitations section that most studies are characterized by small sample sizes, single-center designs and retrospective analyses, all factors that may contribute to heterogeneity and may limit the generalizability of the findings. By explicitly acknowledging these aspects, we aim to provide a balanced interpretation of the evidence while highlighting areas where further high-quality, multicenter, prospective studies are needed.
The manuscript does a good job of describing individual biomarkers (ctDNA, CTCs, EVs, etc.) and their respective strengths. However, a more direct, comparative analysis would be beneficial and prove insightful.
R: We thank the reviewer for this helpful suggestion. Although our manuscript was conceived as a descriptive narrative review, we agree that a concise comparative assessment of circulating biomarkers would increase clarity and highlight their distinct clinical roles. In accordance with this feedback, we have added a short comparative paragraph in the Limitations section summarizing the relative strengths, limitations and areas of applicability of ctDNA, CTCs, EVs, and RNA-based biomarkers.
Table 1 provides a very useful comparison of ctDNA detection technologies (qPCR, ddPCR, NGS), which is a strong point of the paper. However, this is not consistently applied to other biomarkers.
R: Thank you for the suggestion. We agree that extending the comparative approach to other biomarkers would strengthen the manuscript. However, unlike ctDNA—where qPCR, ddPCR and NGS are well-established and directly comparable—the detection methods for CTCs, EVs and RNA-based biomarkers remain highly heterogeneous and lack standardized performance metrics, making a unified table less reliable. Nevertheless, we have now added a concise comparative summary of the main detection techniques used for these biomarkers to provide greater methodological consistency across sections.
Minor comments
Figure 1 attempts to visually conceptualize the sources and applications of liquid biopsy. The figure is somewhat simplistic and does not fully capture the dynamic and complementary nature of the biomarkers discussed in the text
R: Thank you for your comment. To improve the clarity and completeness of the visual summary, I have replaced the original figure with an alternative one taken from a different open-access article published under a Creative Commons Attribution (CC BY) license. This allows the figure to be reused legally while providing a more accurate representation of the sources and applications of liquid biopsy biomarkers.
There are minor shifts between present and past tense when describing study findings.
R: We acknowledge the minor shifts between present and past tense in the description of study findings and have revised the manuscript to ensure consistent use of tense throughout
Reviewer 3 Report
Comments and Suggestions for Authors
This is a very meticulous and comprehensive literature review with 161 references. It is rare that I recommend accepting a manuscript as submitted, but this is one of those cases. I commend the authors for their thorough and diligent work.
Author Response
This is a very meticulous and comprehensive literature review with 161 references. It is rare that I recommend accepting a manuscript as submitted, but this is one of those cases. I commend the authors for their thorough and diligent work.
R: On behalf of all authors, I sincerely thank the reviewer for the positive feedback.
Reviewer 4 Report
Comments and Suggestions for Authors
This paper reviews the advancements in the application of liquid biopsy for diagnosing and treating head and neck cancer. It highlights that head and neck squamous cell carcinoma (HNSCC), ranking as the sixth most prevalent malignant tumor worldwide, has an annual incidence of approximately 600,000 new cases. Early diagnosis is challenging due to the absence of screening programs and the heterogeneity of the tumor, which contributes to suboptimal survival rates. Liquid biopsy offers a novel method for non-invasive tumor monitoring by detecting circulating tumor DNA (ctDNA), circulating tumor cells (CTCs), cell-free RNA (cfRNA), microRNAs (miRNAs), extracellular vesicles (EVs), and viral nucleic acids. Technologically, droplet digital PCR (ddPCR) and next-generation sequencing (NGS) significantly enhance detection sensitivity and specificity, facilitating applications in early diagnosis, risk stratification, prognostic evaluation, and therapeutic monitoring. Specifically, ddPCR is ideal for real-time monitoring and minimal residual disease detection, while NGS is better suited for comprehensive genomic analysis. Saliva samples yield a 100% detection rate for oral tumor ctDNA, whereas plasma is more effective for deep head and neck tumors. Challenges include standardizing tests, establishing sensitivity thresholds, and addressing cost-effectiveness concerns. Future prospective multicenter studies are essential to validate clinical utility and establish standardized analytical workflows. This technology shows great potential for advancing precision medicine in managing head and neck cancer, thereby supporting individualized treatment strategies. It is recommended that the author revise the manuscript in accordance with the following comments.
- It is recommended to briefly mention the main research findings in the abstract, rather than limiting it to a description of methods.
- It is recommended that the author allocate additional space in the introduction section to provide comprehensive background information on head and neck cancer.
- The following references are highly relevant to the author's topic and are recommended for citation.
[1] J. Li, A. Gu, N. Tang, G. Zengin, M.-Y. Li, Y. Liu, Patient-derived xenograft models in pan-cancer: From bench to clinic. Interdiscip. Med. 2025, 3, e20250016. DOI: 10.1002/INMD.20250016
[2] Sharma R, Malviya R. Modifying the electrical, optical, and magnetic properties of cancer cells: a comprehensive approach for cancer management. Med Adv. 2024; 2(1): 3–19. https://doi.org/10.1002/med4.51
[3] Zhang Y, Peng Y, Lin B, Yang S, Deng F, Yang X, Li A, Xia W, Gao C, Lei S, Liao W, Zeng Q. Non-coding RNA and drug resistance in head and neck cancer. Cancer Drug Resist. 2024;7:34. http://dx.doi.org/10.20517/cdr.2024.59
- To address issues related to standardization, sensitivity, and cost-effectiveness, specific solutions or recommendations are proposed, such as promoting the development of standardized protocols for detection methods and conducting large-scale clinical trials for validation.
- The conclusion section presents clear research questions or hypotheses, including what types of studies are required to validate the clinical utility of liquid biopsy, or which biomarkers hold the greatest promise.
- The advantages of combining circulating tumor HPV DNA (ctHPVDNA) with imaging modalities in postoperative surveillance should be emphasized. Additionally, supplementing data on the application of EBV DNA in non-endemic regions is necessary to enhance its global applicability.
Author Response
It is recommended to briefly mention the main research findings in the abstract, rather than limiting it to a description of methods.
R: We thank the reviewer for this helpful suggestion. In accordance with the comment, we have revised the abstract to include the main research findings of the review, summarizing the most relevant evidence on the diagnostic, prognostic, and predictive value of circulating biomarkers in HNSCC.
It is recommended that the author allocate additional space in the introduction section to provide comprehensive background information on head and neck cancer.
R: We thank the reviewer for the comment. We would like to point out that the introduction already includes an overview of head and neck cancer epidemiology, major risk factors and the role of HPV, followed by a section outlining the clinical challenges, including the need for early diagnosis, disease monitoring and improved risk stratification. This provides the context for the relevance of circulating biomarkers, while keeping the focus of the manuscript on the role of liquid biopsy rather than a general review of head and neck cancer.
The following references are highly relevant to the author's topic and are recommended for citation.
R: We thank the reviewer for the comment. The suggested references have been included in the text.
To address issues related to standardization, sensitivity, and cost-effectiveness, specific solutions or recommendations are proposed, such as promoting the development of standardized protocols for detection methods and conducting large-scale clinical trials for validation.
R: We thank the reviewer for the comment. We agree that standardization, sensitivity and cost-effectiveness are key issues in the clinical implementation of liquid biopsy. As noted in the revised manuscript, we have already discussed these aspects in detail and highlighted concrete solutions currently pursued in the field. In particular, we have described ongoing efforts such as the BLOODPAC Consortium and other collaborative initiatives that are actively developing standardized pre-analytical and analytical protocols, reference materials and shared bioinformatics pipelines. Moreover, we comment on the need for large-scale prospective clinical trials to validate assay performance and improve cost-effectiveness. These additions aim to address the reviewer’s concern while maintaining the scope and focus of the review on circulating biomarkers in head and neck cancer.
The conclusion section presents clear research questions or hypotheses, including what types of studies are required to validate the clinical utility of liquid biopsy, or which biomarkers hold the greatest promise.
R: We thank the reviewer for this insightful observation. We agree on the importance of outlining future research directions; in this regard, the current conclusion section already highlights the major gaps that need to be addressed for the clinical translation of liquid biopsy in HNSCC, including the need for standardized workflows, prospective multi-institutional validation studies and improved assay harmonization. Moreover, the conclusion identifies ctDNA—particularly viral ctDNA in HPV-positive disease—as the most mature biomarker, while emphasizing that additional biomarkers (CTCs, EVs, cfRNA, miRNAs) remain at earlier stages of development and require further investigation. In light of this, we believe that the essential research priorities and the biomarkers with the greatest potential are already clearly reflected in the existing conclusion.
The advantages of combining circulating tumor HPV DNA (ctHPVDNA) with imaging modalities in postoperative surveillance should be emphasized. Additionally, supplementing data on the application of EBV DNA in non-endemic regions is necessary to enhance its global applicability.
R: We thank the reviewer for this valuable comment. We agree that the integration of ctHPVDNA with imaging modalities represents an important advance in postoperative surveillance. In the revised manuscript, we have further emphasized this point by highlighting how ctHPVDNA can complement PET-CT and MRI, particularly in cases of indeterminate or ambiguous imaging findings. As detailed in the discussion, several studies demonstrate that ctHPVDNA can detect recurrence earlier than imaging, provides a higher negative predictive value and may help reduce unnecessary procedures such as neck dissection. We have also underlined the synergistic value of combining metabolic imaging with ctHPVDNA kinetics to improve risk stratification and guide early clinical intervention. We agree that discussing evidence from non-endemic regions is important to broaden the global relevance of cfEBV DNA. Accordingly, we have expanded the section on nasopharyngeal carcinoma to include data from studies conducted in non-endemic settings, which demonstrate that cfEBV DNA retains clinical utility also in low-incidence populations. These studies support its role in detecting minimal residual disease, monitoring treatment response, and predicting recurrence even outside endemic areas, reinforcing the biomarker’s applicability across diverse clinical contexts. The manuscript has been updated to reflect this broader perspective.
Round 2
Reviewer 1 Report
Comments and Suggestions for Authors
In their revised review article, the authors have adequately addressed my comments, which in my opinion has improved the content. Therefore, the article is suitable for publication.